# BMIM-BF$_4$ RTIL: Synthesis, Characterization and Performance Evaluation for Electrochemical CO$_2$ Reduction to CO over Sn and MoSi$_2$ Cathodes

Ibram Ganesh 

Centre of Excellence for Artificial Photosynthesis, International Advanced Research Centre for Powder Metallurgy and New Materials (ARCI), Balapur Post, Hyderabad 500 005, India; ibramganesh@arci.res.in or dribramganesh@gmail.com; Tel.: +91-40-24452396

**Abstract:** Development of a practicable artificial photosynthesis process has been considered today as one of the top-most research priorities to address the problems related to the global warming and the associated social cost of carbon, and to develop the renewable fuels employable in place of fossil fuels. For this purpose, a simple and inexpensive route has been devised to synthesize a high-purity bmim[BF$_4$] to employ as a helper catalyst to promote the electrochemical CO$_2$ reduction (ECR) to CO formation over Sn and MoSi$_2$ cathodes. The rates of CO$_2$ reduction over Sn and MoSi$_2$ cathodes have been determined to be >110 mA/cm$^2$ during cyclic voltammetry. The CO formation at a current density of >100 mA/cm$^2$ in the ECR reaction is very essential to consider this reaction for industrial practice when the required electricity derived from sunlight is available at an affordable price. The bmim[BF$_4$] mediated ECR reaction over Sn and MoSi$_2$ cathodes has been identified to be a revere reaction of CO oxidation in air. The experiments with isotopic $^{13}$CO$_2$ confirmed that CO$_2$ is the only source of CO formation in the ECR reaction. The underlying reaction mechanism in bmim[BF$_4$] mediated ECR reaction over Sn has been presented and discussed in this article.

**Keywords:** electrochemical CO$_2$ reduction; ionic liquid; bmim-BF$_4$; Sn; MoSi$_2$; isotopic $^{13}$C-CO$_2$; NaHCO$_3$

## 1. Introduction

Today, the electrochemical carbon dioxide (CO$_2$) reduction (ECR) to fuel chemicals, such as, CO, methanol, etc., using electricity derived from sunlight, which is popularly known as the artificial photosynthesis (AP) process, is considered as one of the top most research priorities to address the problems related to the (i) CO$_2$ associated global warming, and the associated social cost of carbon, (ii) to store the renewable and surplus energy in high energy density liquid fuels employable directly in place of petrol (gasoline) and diesel in the present existing energy distribution infrastructure (i.e., internal combustion (IC) engines) so that there would not be any severe economical consequences while transforming from fossil fuel energy dependency to non-fossil fuel, renewable and solar energy dependency and (iii) to arrest the depletion of fossil fuels, mainly the important resource of crude oil required for future generations [1–10]. Furthermore, today, >80% of the global primary energy requirement is met from fossil fuels, and this is the reason for the increased concentration of CO$_2$ in the atmosphere to the extent of >400 ppm today [11]. This anthropogenic atmospheric CO$_2$ reacts with seawater when it comes into contact with the surface of seawater, which is spread across more than 70% of the earth surface, at an hydration equilibrium constant (Kr) of $1.7 \times 10^{-3}$ thereby leading to the formation of bicarbonates and carbonates of calcium, sodium and magnesium in the seawater as per the reactions shown in Equations (1)–(9) while releasing two protons (H$^+$) for every CO$_2$ molecule reacted

with water into the seawater thus contributing to the acidification of seawater [1–13]. This latter process is popularly called "ocean acidification", which is responsible for the formation of biological calcium carbonate minerals [14]. As of today, the readily available technology to address the problems related to the $CO_2$ associated global warming and the social cost of carbon, and to avoid the acidification of seawater, is the "$CO_2$ sequestration", which is also called "carbon capture and storage (CCS)" [12,13]. It is known that when $CO_2$ is pumped into deep seawater below 3500 m from top of the seawater surface as a part of the CCS process, a stable $CO_2$ clathrate (Equation (10)), is formed, which is a snow-like crystalline substance composed of $H_2O$ ice and $CO_2$, under the pressure of >350 bar and temperature of 3 °C, which are present at below 3500 m deep seawater [13]. Recently, Matter et al. [12] reported a new $CO_2$ sequestration process. In this latter process, when a mixture of water and $CO_2$ is pumped 540 m deep into the Iceland rocks, this acidic solution causes the leaching of the Mg and Ca metals out of the basalt rocks and converts them into $MgCO_3$ and $CaCO_3$ rocks. These latter rocks have been found to be stable for more than a two-year period. The cost of this new $CO_2$ sequestration process has been estimated to be about 18 USD for disposing off a ton of $CO_2$ gas.

$$CO_{2(aq)} + H_2O_{(aq)} \leftrightarrows H_2CO_{3(aq)} \ (K_r = 1.7 \times 10^{-3}) \tag{1}$$

$$[CO_{2(aq)}] \leftrightarrows H^+ + HCO_3^- \ ([H^+][HCO_3^-] = K_{a1}[CO_{2(aq)}]) \ (pK_{a1} = 6.363 \text{ at } 25\,°C \text{ and } I = 0) \tag{2}$$

$$HCO_3^- \leftrightarrows H^+ + CO_3^{2-} \ ([H^+][CO_3^{2-}] = K_{a2}[HCO_3^-]) \ (pK_{a2} = 10.329 \text{ at } 25\,°C \text{ and } I = 0) \tag{3}$$

$$HCO_3^- + Na^+ \leftrightarrows NaHCO_3 \tag{4}$$

$$CO_3^{2-} + 2Na^+ \leftrightarrows Na_2CO_3 \tag{5}$$

$$CO_3^{2-} + Mg^{2+} \leftrightarrows MgCO_3 \tag{6}$$

$$CO_3^{2-} + Ca^{2+} \leftrightarrows CaCO_3 \tag{7}$$

$$2HCO_3^- + Mg^{2+} \leftrightarrows Mg(HCO_3)_2 \tag{8}$$

$$2HCO_3^- + Ca^{2+} \leftrightarrows Ca(HCO_3)_2 \tag{9}$$

$$CO_2 + 8H_2O \leftrightarrows CO_2 \cdot 8H_2O \ (<10\,°C \text{ and } >45 \text{ bar}) \tag{10}$$

Although, at present, the $CO_2$ sequestration into the deep seawater is the only available option to address the $CO_2$ associated global warming problem, the conversion of $CO_2$ into fuel and other value added chemicals by following the carbon capture and utilization (CCU) process can be considered a more beneficial one [13–28]. The CCS process apart from being expensive (when $CO_2$ is transported from far distances) and laborious, it irreversibly blocks the important resource of $C_1$ carbon in the form of $CO_2$ clathrate in deep seawater, which cannot be used for any of the human beneficial activities. In fact, the $CO_2$ shall be participated in the natural carbon cycle that is required for the sustainability of life on the earth surface. Furthermore, the $CO_2$ to be sequestered is normally captured at major outlets such as thermal power plants, cement industry, steel factories, gas refineries, etc., which are situated most often far away from those sites used for sequestration [15,16]. In certain places like those in Europe, transportation in pipelines to the storage sites is quite difficult, and it will increase the cost of the process by at least 15–20%, which is considered to be unacceptable. The Department of Energy (DOE), USA, suggests that the transportation of $CO_2$ in tankers on the road is not acceptable if the distance is more than 100 km. In the total CCS process, about 35–40% is the transportation and storage costs. This cost would be between 35 and 50 €/ton for the early commercial phase (after year 2020) and between 60 and 90 €/ton during the demonstration phase, when transportation of $CO_2$ is made by pipelines for distances not over 200–300 km. This cost would be further increased if the transportation was made by the road [15,16]. Of course, it can be well-understood that the CCU process is meaningful only if the energy required for driving the thermodynamic energy consuming $CO_2$ conversion to fuel chemicals is obtained from any of the available renewable energy resources

at an affordable cost. In addition to that, today, none of the existing industrial $CO_2$ conversion processes has the ability to consume all of the $CO_2$ gas generated at all major outlets across the globe as the consumption of products formed in these processes is limited in the present society. Furthermore, these industrial $CO_2$ conversion processes cannot store any external energy in the form of chemical bonds of products formed in them as these reactions proceed by utilizing the chemical potentials of the reactants as they are exothermic, stoichiometric, neutralization, spontaneous and energy releasing ones in nature [1,8,10,15,16,19–23]. Whereas, energy storing $CO_2$ conversion processes such as electrochemical $CO_2$ reduction (ECR) to CO, etc., are yet to be fully developed to practice at industry for commercial purposes [8–10,15,23].

Since, $\Delta G°$ values of $CO_2$ reduction to CO and methanol [8,17] are positive ($\Delta G = -RT \ln k$ and $\Delta G = \Delta H - T\Delta S$; where R: universal gas constant (8.314 J/mol K); T: temperature in Kelvin; H: enthalpy (J/mol) and S: entropy (J/K)), these reduction reactions can store external energy in the products formed. Furthermore, the $E^0$ of one electron reduced $CO_2$ to $CO_2^{\bullet-}$ intermediate formation is about $-1.90$ V vs. NHE, whereas, the $E^0$ of two-electrons reduced CO formation with two-protons ($H^+$) and two-electrons ($e^-$) is only about $-0.17$ V vs. NHE [8,17,23]. These $E^0$ values suggest that a catalytic system is required to bypass one-electron reduced $CO_2^{\bullet-}$ intermediate formation in the ECR to CO formation reaction so that the wastage of at least 1.73 V as an overpotential for every CO molecule formed can be avoided. The CO gas formed in the ECR reaction, and the $H_2$ obtained from water in an electrolyzer can be employed to synthesize methanol, which in turn can be converted into a synthetic petrol (gasoline) by following an industrially practiced methanol-to-gasoline (MTG) process [1,23]. The commercial methanol is produced at a pressure of $p = 50$–100 bar and temperature of 200–300 °C over the $Cu/ZnO/Al_2O_3$ catalyst [1,18]. The cathodic systems such as, Sn, Au, Ag, Bi, $MoS_2$, etc., employed in conjunction with an imidazolium-based room temperature ionic liquid (RTIL) helper catalysts such as 1-butyl-3-methylimidazolium tetraflouroborate (bmim[$BF_4$]) have been identified to be catalytic systems capable of bypassing the formation of a single-electron reduced $CO_2^{\bullet-}$ intermediate in the ECR reaction to form CO at much lower overpotentials with considerably high selectivity and product formation rate [2–5,23–28]. Among various inexpensive base metal cathodes employed so far for performing the ECR to CO formation reaction, only Sn in conjunction with bmim[$BF_4$] has been identified to be a simple, and very efficient electrocatalytic system. Recently, the relatively inexpensive Mo [29], $MoS_2$ [2], $MoO_2$ [30] and $Mo_2CTx$ MXenes (MXenes are a family of two-dimensional transition metal carbide/nitride materials with metallic-like conductivity, and with a general formula of $M_{n+1}X_nT_x$, where M represents an early transition metal, X is carbon and/or nitrogen, with n in the range of 1–4 and $T_x$ represents surface termination groups including –O, –F, etc.) [31] in comparison to noble metal cathodes have also been found to be excellent electrocatalytic systems to perform ECR to form methanol, CO, etc., under different reaction conditions. However, in comparison to Mo, the $MoS_2$ has exhibited higher activity for CO formation when it was employed in conjunction with bmim[$BF_4$] RTIL in terms of the product selectivity, Faradaic efficiency (FE) and overpotential required. However, the $MoS_2$ deposited by following chemical vapor deposition on a conductive carbon surface from Mo and S cannot be as robust as the Mo metal electrode in terms of physical integrity [2]. At high turbulent conditions, which are prevalent during industrial practice of the ECR reaction at pressures exceeding >50 bar, the $MoS_2$ can be peeled off from the surface of electrically conductive carbon substrate. The molybdenum disilicide, $MoSi_2$, is another compound popularly known as the Kanthal® Super heating element to the materials scientists. It is a unique material combining the best properties of metallic and ceramic materials, and can be synthesized by following an expensive self-propagating high-temperature synthesis (SHS) route using Mo and Si starting powders [32], and it can be fully densified into strong objects in different shapes by introducing a considerable amount of porosity in them by following the conventional powder processing and sintering routes. Like, a metallic material, $MoSi_2$ has good heat and electrical conductivity, and like ceramics, it withstands corrosion and oxidation, and has low thermal expansion. It can withstand about 1900 °C in the open-air atmosphere. In this study, for the first time, $MoSi_2$ was investigated as a

cathode in conjunction with bmim[$BF_4$] RTIL to perform the ECR reaction in a non-aqueous aprotic catholyte solution separated by a nafion membrane from an anolyte solution of 0.5 M sodium phosphate (pH = 7.4) buffer containing 0.5 mM $Co^{2+}$ ions and compared its electrocatalytic performance with the one obtained over the Sn cathode under identical reaction conditions [23,28].

Nevertheless, by involving very expensive bmim[$BF_4$] RTIL as a helper catalyst, it is not possible to devise an inexpensive ECR process to produce CO for commercial purposes. Since, none of the reported methods in the literature are straight forward to synthesize high-purity colorless bmim[$BF_4$] RTIL from locally available inexpensive starting raw materials (1-bromobutane (BB) and 1-methylimidazole (MI)) as summarized in Table 1 [33–37], in this article, a simple and straight forward synthetic route is reported. In this new route, the water employed to decolorize the bmim[Br] intermediate formed out of BB and MI starting reactants with the help of activated charcoal was removed by following the freeze drying technique for the first time, which is known for selectively removing the water from high-boiling point organic reaction mixtures at relatively faster rates.

**Table 1.** The procedures reported for the synthesis of high-purity bmim[$BF_4$], Bmim[$PF_6$], bmim[Br] and bmim[Cl] from 1-methylimidazole (MI), 1-bromobutatne (BB), 1-chlorobutatne (CB), $KBF_4$, $KPF_6$ and $NaBF_4$ reactants [†].

| IL Synthesized | Reactants | Reaction Procedure | Ref. |
|---|---|---|---|
| Bmim[Br] | MI and BB | Refluxing reactants mixture (RM) at 70 °C for 48 h | [33] |
| Bmim[Br] | MI and BB | Refluxing RM at <40 °C for 24 h | [34] |
| Bmim[Cl] | MI and CB | Refluxing RM at 70 °C for 7 days | [35] |
| Bmim[Cl] | MI and CB | Refluxing RM in acetonitrile (MeCN) at 80 °C for 48 h | [35,36] |
| Bmim[$BF_4$] | Bmim[Br] and $NaBF_4$ | Refluxing RM in acetone at 40 °C for 10 h and extraction with DCM | [33] |
| Bmim[$BF_4$] | Bmim[Cl] and $KBF_4$ | Stirred RM in $H_2O$ at RT [†] for 2 h and vacuum separation with DCM | [35,36] |
| Bmim[$BF_4$] | Bmim[Cl] and $NaBF_4$ | Stirred RM in $H_2O$ at 45 °C for 15 min, and distillation with DCM | [37] |
| Bmim[$BF_4$] | Bmim[Br] and $NaBF_4$ | Refluxing RM at <25 °C for 3 h and continuous liquid–liquid extraction with DCM | [34] |
| Bmim[$PF_6$] | Bmim[Cl] and $KPF_6$ | Stirred RM in $H_2O$ at RT for 2 h and vacuum separation with DCM | [35] |

[†] RT—room temperature.

Furthermore, the reaction conditions needed to extract bmim[$BF_4$] RTIL by following a continuous liquid–liquid extraction technique from the reaction mixture containing inorganic byproducts such as, NaBr, KBr, etc., using dichloromethane (DCM) as an extraction solvent is not reported clearly in the existing literature.

In view of the above, the main objective of the present study was to devise a simple and straight forward route for the synthesis of high-purity bmim[$BF_4$] using locally available inexpensive starting raw materials. Accordingly, an high-purity bmim[$BF_4$] was synthesized and was thoroughly characterized by means of [1]H- and [13]C-NMR, FT-IR, mass spectral and inductively coupled plasma-optical emission spectrometry (ICP-OES ) techniques, and finally evaluated its efficacy as an helper catalyst (i.e., as a mediator) to promote the ECR to CO formation reaction over Sn as well as $MoSi_2$ bulk monolith cathodes as this latter reaction is a part of an important AP process. For comparison purposes, a commercially available high-purity (≥97.0%) bmim[$BF_4$] RTIL procured from Aldrich, St. Louis, MO, USA was also utilized in this investigation.

## 2. Materials and Methods

### 2.1. Synthesis of 1-Butyl-3-methylimidazolium Tetraflouroborate (bmim[$BF_4$])

The detailed procedure involved in the synthesis of bmim[$BF_4$] RTIL is schematically shown in Scheme 1. In a typical experiment, 27 mL n-bromobutane (BB; 98% purity, stabilized with silver wire, Sisco Research Laboratories (SRL), Mumbai, India) was slowly introduced into 11 mL freshly distilled 1-methylimidazole (MI; Tokyo Chemical Industry (TCI), Tokyo, Japan) solution taken in a 500 mL 2-neck round bottom (RB) flask fitted with a dropping (i.e., equalization) funnel and stirred for >48 h while maintaining the reaction temperature below 40 °C. The ratio maintained between MI

and BB was 1:1.5, respectively [33]. The resultant 1-butyl-3-methylimidazolium bromide (bmim[Br]) yellowish-white solid (Figure 1a) was washed thrice with excess amount of ethyl acetate (EA; 98.0% pure, SQ, Fisher-Scientific, Qualigens, Mumbai, India; each time with 200 mL) to remove the unreacted starting raw materials as bmim[Br] does not dissolve in EA. Finally, the decolorization of bmim[Br] was accomplished by dissolving it in about 250 mL of deionized water taken in a 1000 mL borosil glass beaker, to which about 3–5 g of activated charcoal (decolorizing agent; activated carbon, 400 superior extra pure, AR grade, SRL, Mumbai, India) was introduced and then refluxed at 100 °C with vigorous stirring for >24 h. The activated charcoal was then filtered off using a Buchner funnel containing two numbers of 125 mm diameter (φ) Whatman filter papers (grade 41 and 42, Whatman™, Ashless GE Healthcare Life Science, Marlborough, MA, USA) to obtain a very clear transparent solution, which was then sprayed into liquid $N_2$ present in a 1000 mL vacuum (Dewar) flask. The resultant transparent solid was then dried on a freeze drying machine overnight to become a clear transparent solution. Thus, obtained solution was further subjected to a rotary evaporation at 80 °C, and then placed in a vacuum oven (Lindberg Blue, 3M, St. Paul, MN, USA) maintaining at 80 °C overnight to obtain white colored pure bmim[Br] solid (Figure 1b).

**Scheme 1.** Schematic representation of various steps involved in a typical procedure employed for the synthesis of high-purity bmim[BF$_4$] room temperature ionic liquid (RTIL).

The bmim[BF$_4$] RTIL was prepared by performing a metathesis reaction between the above-synthesized bmim[Br] solid, and sodium tetrafluoroborate (NaBF$_4$; 98.5%, SRL, Mumbai, India). In a typical experiment, about 25 g bmim[Br] was dissolved in about 250 mL deionized water, which was then added with 13 g NaBF$_4$ (slightly excess than the stoichiometric requirement). The resultant solution was vigorously stirred at room temperature for >24 h, and washed with diethyl ether (DEE) using a separating funnel leaving the sodium bromide (NaBr; a byproduct formed in this reaction; confirmed by XRD analysis, Figure S1) in the water layer [33]. Then, the DEE was evaporated

on a rota-evaporator to obtain pure bmim[BF$_4$] RTIL, which was then dried at 70 °C overnight in a vacuum oven (Lindberg Blue, St. Paul, MN, USA). Alternatively, the clear aqueous solution mixture of bmim[Br] and NaBF$_4$ (13 g) in 250 mL deionized water was also stirred at room temperature (RT) for 24 h, and then transferred into a continuous liquid–liquid extraction apparatus (Figure S2) and extracted bmim[BF$_4$] with dichloromethane (DCM; anhydrous, ≥99.8%, contains 40–150 ppm amylene as a stabilizer, Sigma-Aldrich) solvent for about 48 h. The DCM solution was then filtered through a plug of silica (100 g), and removed it on a rota-evaporator to obtain a high-purity bmim[BF$_4$] RTIL with >95% yield. Henceforth, the bmim[BF$_4$] synthesized in this study is referred to as ARCI-bmim[BF$_4$], and the one procured from Aldrich, USA is referred to as Aldrich-bmim[BF$_4$].

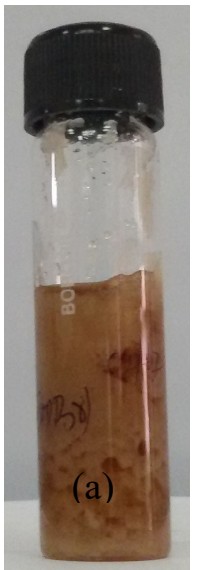
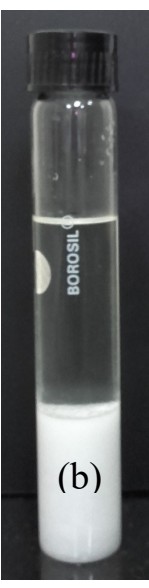

**Figure 1.** The digital photographs of the as synthesized bmim[Br] compound before (**a**) and after (**b**) washing with the activated charcoal to remove the fluorescent colored impurities.

*2.2. Electrochemical Experiments*

ECR reactions were performed using controlled potential bulk electrolysis (CPBE) as well as cyclic voltammetry (CV) techniques using a three-electrode based Metrohm (PGSTAT 302N, Autolab B.V., Kanaalweg 29G, 3526 KM, Utrecht, The Netherlands) electrochemical workstation. The potentials in non-aqueous aprotic catholyte solution were measured using Ag/0.01 M Ag$^+$ (AgNO$_3$, 99.1% pure, SQ, Fisher-Scientific, Mumbai, India), and thus obtained potentials are then converted into NHE (normal hydrogen electrode) using Equation (11) [23,28,38,39].

$$E(NHE) = E(Ag/(0.01 \text{ M Ag}^+) + 0.548 \text{ V} \tag{11}$$

The reference electrode (Ag/0.01 M Ag$^+$) was calibrated by following a reported procedure as elaborated in the supporting information file (Figure S3) [40]. The electrolyte solution prepared by mixing high-purity and fully dried MeCN (Chromosol V$^®$ Plus, for HPLC, >99.9%, Sigma-Aldrich), St. Louis, MO, USA), n-Bu$_4$NPF$_6$ (0.1 M, ≥99.0%, Sigma-Aldrich) and bmim-BF$_4$ (20 mM; synthesized in this study) was employed in the CV experiment. Prior to the experiment, for about 30–45 min, the electrolyte solution was purged with either Ar (>99.9%, Production Needs Gases (Pvt) Ltd., Hyderabad, India) or $^{12}$CO$_2$ (99.99%, M/s. Sicgil India (Pvt) Ltd., Chennai, India) at 50 sccm to remove gaseous impurities if any are present while vigorously stirring it on a magnetic stirrer in a single-compartment electrochemical (EC) cell depending on the nature of the reaction to be carried out. The CV recordings were made at a scanning rate of 10–100 mV/s [23,28,38]. While recording CV, the gas being sent into the electrolyte solution was used as a blanket on the headspace of the

electrolyte solution at a pressure slightly excessive than that of the atmospheric one so that no gas impurities such as, oxygen, moisture, etc., can enter into the electrolyte solution from the environment. The MeCN drying was performed by storing it over anhydrous sodium carbonate ($Na_2CO_3$, ≥99.0%, Sigma-Aldrich, St. Louis, MO, USA) for more than a week while stirring it occasionally. After that, it was twice subjected to a fractional distillation operation under reduced Ar pressure before storing it in a dark colored bottle under Ar gas. The residual moisture from MeCN was removed by passing it through a column of $Al_2O_3$ (grade 1, Merck) treated under vacuum at 360 °C. In order to compensate the evaporation of MeCN from the electrolyte solution during purging with the gas, the gas was passed through MeCN placed in a 500 mL volume glass-trap apparatus. The surface activation or regeneration, or purification of metal electrodes was accomplished by polishing them using an $Al_2O_3$ powder (an average particle size of 0.05 µm) slurry prepared using Millipore (ultrahigh-purity) water, and the surface cleaning of these polished metal electrodes was achieved by performing a sonication operation for 5 min in ultra-high purity Millipore water [23,28,38]. The geometric area of Sn and $MoSi_2$ working electrodes (WEs) was about <2 square millimeters, and a Pt wire with a diameter of about 0.5 mm (ϕ) procured from M/s. BASi, USA, was employed as auxiliary or counter electrode (AE or CE). The electrode Sn (>99.8%) metal was supplied by M/s. Tirupati Metals, Fathenagar, Hyderabad, whereas, the $MoSi_2$ was a centre cut-portion of used Kanthal Super 1900 heating element [41].

The CPBE experiments of ECR was performed in an all-glass gas-tight two-compartment electrochemical cell separated by a nafion (Alfa Aesar, Nafion® N-1178, 0.180 mm thick, ≥0.90 meq/g exchange capacity, $R_f[OCF_2CF(CF_3)_2]_nOCF_2CF_2SO_3H$, 15 cm × 15 cm) membrane at room temperature at a reduction potential of −1.952 V vs. NHE [23,28,38]. The catholyte solution employed was a mixture of MeCN + n-$Bu_4NPF_6$ (0.1 M) + bmim[$BF_4$] (50 mM; either in house synthesized or the one procured from Aldrich, St. Louis, MO, USA with a purity of ≥97.0%; 91508-5G) saturated or/and continuously purged with $^{12}CO_2$ or $^{13}CO_2$ (99 atom% $^{13}C$, <3 atom% $^{18}O$, $^{13}C$ labeled carbon dioxide $^{13}CO_2$, 364592-10L, Sigma-Aldrich, St Louis, MO, USA) gas at a rate of 50 sccm [28]. An anolyte solution employed was a buffer (pH = 7.4) solution of sodium phosphate (NaPi; 0.5 M) containing $Co^{2+}$ ions (0.5 mM; cobaltous nitrate, $Co(NO_3)_2 \cdot 6H_2O$, 97–100% pure, SQ, Fisher-Scientific, Qualigens, Mumbai, India). Refer supporting information (SI) file to know the procedure employed to prepare 0.5 M NaPi buffer (pH = 7.4) solution. Similar to the CV experiments, the catholyte solution employed in the CPBE was also purified by purging the gas prior to the experiment. The active geometric areas of cathode (a cut portion of Sn metal sheet, and $MoSi_2$ 8 mm diameter cylindrical shape rod that was vertically cut into two-layers along the length of the rod using an Isomet 1000 Precision Cutter, BUEHLER, Stuttgart, Germany) and anode (Pt foil) employed were 1.5 $cm^2$ and 4–5 $cm^2$, respectively [28]. During the experiments, about 0.1 mL of headspace gas was withdrawn into a deaerated gas-tight glass-syringe (Hamilton make) through a rubber septum (Suba-seal septa, Aldrich), and was injected into gas chromatography (GC; Agilent Technologies, 7890A, GC system, G3440A, Serial # CN10521016)–mass spectrometry (MS; Agilent Technologies, 5975C, inert XL EI/CI MSD with Triple-Axis Detector, G3174A, Serial # US10494610) equipment fitted with a capillary column (0.530 mm × 30 m, 50 micron, −60 to 300 °C, HP-MOLSIEVE, 19095P-MSO, Agilent Technologies, Inc., Santa Clara, CA, USA) packed with 5 Å molecular sieves to separate and detect the CO, $H_2$, $O_2$ and $N_2$ gases using the thermal conductivity detection (TCD) method coupled with mass-spectral analysis [28]. Contamination of the headspace by air leak was quantified by determining the $N_2$ present in the headspace areas of both the compartments (using the $N_2$ peak on GC traces). The yield, FE, TOF (turn-over-frequency) and the number of moles of product gases formed are determined as per the procedures described in SI [28].

### 2.3. Material Characterization

$^1H$-NMR spectra were recorded at 300, 400 or 500 MHz, and the $^{13}C$-NMR spectra at 100 or 125 MHz in $CDCl_3$ + DMSO for the determination of molecular structures and conformations. For all NMR analysis, approximately 30 mg of the ARCI-bmim[Br], or Aldrich-bmim[$BF_4$] was added

into a 5 mm NMR tube. $^1$H-NMR data in ppm ($\delta$) from the internal standard (tetramethyl silane, TMS, 0.0 ppm), chemical shift (multiplicity, integration) and the $^{13}$C-NMR data (internal standard CDCl$_3$) in ppm ($\delta$) were reported. The values of $^1$H-NMR spectral lines and the coupling constants are given in hertz (Hz). The mass spectra were recorded by using a high-resolution Q-TOF mass spectrometer. The micrographs were recorded using a scanning electron microscope (SEM; JSM–5410, JEOL, Akishima, Tokyo, Japan) with an energy dispersive scanning (Sigma 3.42 Quaser, Kevex, Mahwah, NJ, USA) attachment for qualitative and quantitative microanalysis. The FT-IR spectrum of bmim[BF$_4$] was recorded using Thermo Nicolet Nexus 670 Fourier Transform Infrared Spectrometer, Waltham, MA, USA, containing the detector of DTGS KB with a KBr beam splitter and the resolution of 4 cm$^{-1}$. The sample preparation method employed was a neat mode. In order to identify whether Na impurity in the form of NaBr present in ARCI-bmim[BF$_4$] and in Aldrich-bmim[BF$_4$], they were analyzed by ICP-OES- using Thermo Scientific™ iCAP™ 7400 ICP-OES, Thermo Fisher Scientific, USA, instrument containing iTEVA Security software. The moisture impurity from bmim[BF$_4$] was removed by placing it in a vacuum ($1 \times 10^{-3}$ torr) over at 80 °C overnight. The ICP-OES measurement parameters employed are as follows: frequency, 27.12 MHz; power, 1.15 kW; demountable quartz torch, Ar/Ar/Ar; coolant gas, Ar, 12.0 L min$^{-1}$; auxiliary gas, Ar, 0.5 L min$^{-1}$; nebulizer gas flow, Ar, 1.0 L min$^{-1}$; nebulizer pressure, 2.4 bar; sample flow rate into glass spray chamber, 0.5 L min$^{-1}$ and wavelength range of the monochromator 165–460 nm. The sample digestion system employed was the Microwave Reaction System (CEM Corporation, Matthews, NC, USA) containing one touch MARS 6 software. For this purpose, the 1 mL bmim[BF$_4$] was mixed with 9 mL HNO$_3$ (65%, Suprapur$^®$, Merck, Germany) and 3 mL HClO$_4$ (perchloric acid, 70%, AR/ACS, Finar, Andheri (E), Mumbai), and was digested using conditions as follows: power: 800 W; ramp: 20 min; holding time: 15 min and temperature: 210 °C.

## 3. Results

### 3.1. Characteristics of Bmim[Br] and Bmim[BF$_4$]

$^1$H- and $^{13}$C-NMR spectra of the ARCI-bmim[Br] white solid compound, ARCI-bmimBF$_4$] and Aldrich-bmim[BF$_4$] colorless RTILs are given in Figures S4a,b, S5a,b and S6a,b, respectively. For clarity purposes, the $^1$H- and $^{13}$C-NMR signal positions ($\delta$ in ppm) and the multiplicity of these signals revealed by [bmim]$^+$ (1-butyl-3-methylimidazolium cation; Scheme 2) of bmim[Br] and bmim[BF$_4$] RTILs are summarized in Table 2.

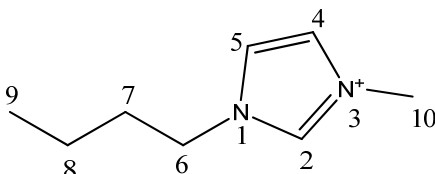

**Scheme 2.** Structure of the 1-butyl-3-methylimidazolium cation [bmim]$^+$.

Furthermore, in order to analyze the multiplicity of $^1$H- and $^{13}$C-NMR signals revealed by ARCI-bmim[BF$_4$], its high resolution spectral images recorded at smaller scales are given in Figure 2a,b and Figure 3, respectively. A closer look into all these six different types of NMR spectra and into the data of Table 2 suggest that ARCI-bmim[Br] and ARCI-bmim[BF$_4$] have the chemical composition of the 1-butyl-3-methylimidazolium (i.e., [bmim]$^+$; Scheme 2) cation. Furthermore, the $^1$H- and $^{13}$C-NMR signals of in-house synthesized bmim[Br] as well as bmim[BF$_4$] match very well with those revealed by Aldrich-bmim[BF$_4$]. Furthermore, these analytical data are in line to those reported in the literature for bmim[BF$_4$] [33]. Furthermore, the signal positions ($\delta$ in ppm) and the multiplicity of various NMR signal shown in Figures 2 and 3 suggest that the ARCI-bmim[BF$_4$] is indeed bmim[BF$_4$] RTIL and it does not contain any other organic impurities in it [33]. The slight difference noted for the NMR signal pertaining to the proton attached to the C$_2$ carbon of imidazolium ring of ARCI-bmim[BF$_4$]

and Aldrich-[BF$_4$] (Table 2—column 3) can be explained as follows: the NMR signal of the proton attached to C$_2$ carbon undergoes shifting that is strongly influenced by the hydrogen bonding with the compounds present around this C$_2$ carbon as it is the most acidic proton compared with the other two protons attached to C$_4$ and C$_5$ carbon atoms of the imidazolium ring [42]. In a study, Chen et al. [43], found that the proton NMR chemical shifts for the C$_2$ carbon of the imidazolium ring of ionic liquids occurs due to a more complex mixture of interionic interactions between the ring and the anion including hydrogen bonding interactions. It can be seen from the Table 2 data of the present study that the bmim[Br]'s C$_2$ proton chemical shift is relatively higher than those of bmim[BF$_4$] RTILs. The presence of small amounts of bmim[Br] as an impurity in the ARCI-bmim[BF$_4$] can result in the observed small chemical shift differences.

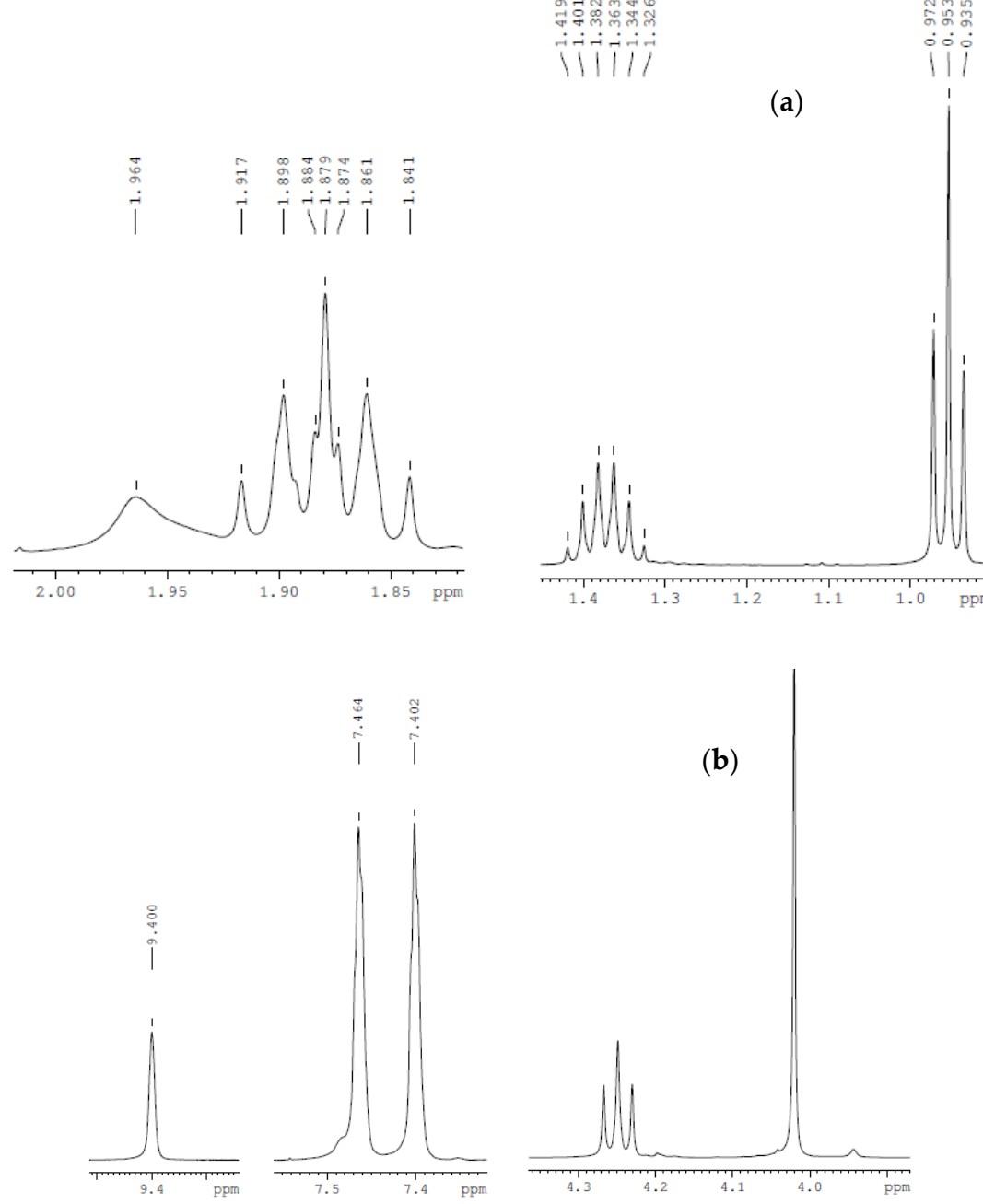

**Figure 2.** (a,b) $^1$H-NMR spectrum of ARCI-bmim[BF$_4$] RTIL.

**Table 2.** $^1$H- and $^{13}$C-NMR spectral data of [bmim]$^+$ group of ARCI-bmim[Br], ARCI-bmim[BF$_4$] and Aldrich-bmim[BF$_4$] $^\dagger$.

| Compound | Analysis | NMR Signal Position δ in Ppm) of [bmim]$^+$ Group $^¥$ (Multiplicity $^\ddagger$) | | | | | | | |
|---|---|---|---|---|---|---|---|---|---|
| | | C$_2$ (1H) | C$_4$ (1H) | C$_5$ (1H) | C$_6$ (2H) | C$_7$ (2H) | C$_8$ (2H) | C$_9$ (3H) | C$_{10}$ (3H) |
| bmim[Br] | $^1$H-NMR | 9.708 (s) | 7.306 (s) | 7.454 (s) | 4.288 (t) | 1.874 (quint) | 1.425 (sext) | 0.958 (t) | 4.063 (s) |
| bmim[BF$_4$] | $^1$H-NMR | 9.40 (s) | 7.40 (s) | 7.46 (s) | 4.25 (t) | 1.87 (quint) | 1.36 (sext) | 0.95 (t) | 4.02 (s) |
| bmim[BF$_4$] $^\ddagger$ | $^1$H-NMR | 8.838 (s) | 7.22 (s) | 7.32 (s) | 4.20 (t) | 1.866 (quint) | 1.36 (sext) | 0.96 (t) | 3.962 (s) |
| bmim[Br] | $^{13}$C-NMR | 137.001 | 121.975 (d) | 123.564 (d) | 49.759 (d) | 32.002 | 19.344 | 13.329 | 36.545 |
| bmim[BF$_4$] | $^{13}$C-NMR | 135.47 | 121.722 (d) | 123.075 (d) | 48.90 (d) | 31.193 | 18.542 | 12.591 | 35.521 |
| bmim[BF$_4$] $^\ddagger$ | $^{13}$C-NMR | 136.59 | 121.931 (d) | 123.514 (d) | 49.875 (d) | 31.865 | 19.374 | 13.294 | 36.315 |

$^\dagger$ Refer to the experimental section to know the recording conditions of $^1$H- and $^{13}$C-NMR spectra. $^\ddagger$ Multiplicities are indicated as follows: s = singlet; br s = broad singlet; d = doublet; dd = doublet of doublet; t = triplet; q = quadruplet; qunit = quintuplet; sext = sextuplet; sept = septuplet and m = multiplet. $^¥$ Eight different carbon atoms of the [bmim]$^+$ group: C$_2$ (1H and C of ring between two N atoms); C$_4$ (1H and C of ring attached to N$_{(2)}$ with methyl group); C$_5$ (1H and C of ring attached to N$_1$ with butyl group); C$_6$ (2H and 1st C of butyl group attached to N$_1$); C$_7$ (2H and 2nd C of butyl group attached to N$_1$); C$_8$ (2H and 3rd C of butyl group attached to N$_{(1)}$); C$_9$ (3H and 4th C of butyl group attached to N$_{(1)}$) and C$_{10}$ (3H and C of methyl group attached to N$_{(2)}$). Aldrich-bmim[BF$_4$] (≥97.0%, 91508-5G) procured from Aldrich, St. Louis, MO, USA.

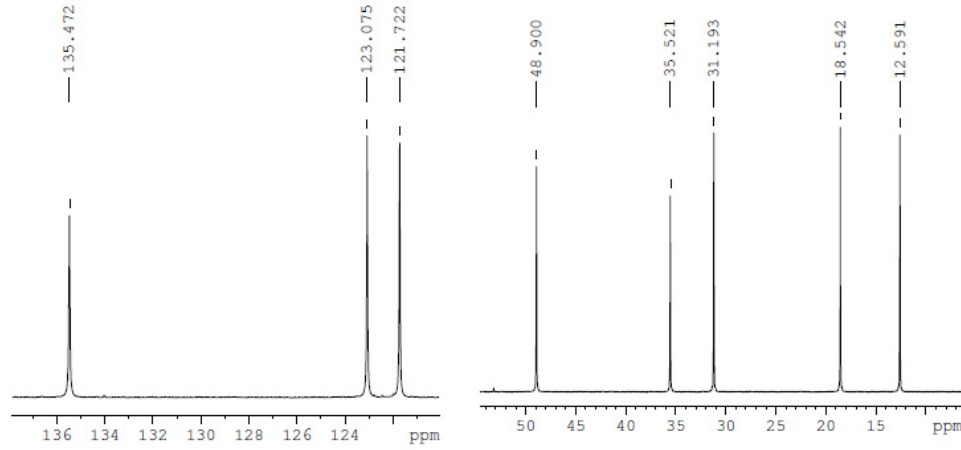

**Figure 3.** $^{13}$C-NMR (125 MHz, CDCl$_3$ + DMSO) spectrum of ARCI-bmim[BF$_4$] RTIL.

Figure 4 presents the FT-IR spectra of both ARCI-bmim[BF$_4$] and Aldrich-bmim[BF$_4$]. In general, it can be seen that both of the RTILs revealed the same kind of FT-IR transmission bands and the difference is only in the percentage of transmission. These results suggest that the functional groups and chemical bonds present in both of these RTILs are the same. The presence of all the functional groups of bmim[BF$_4$] that cause IR bands are seen in the FT-IR spectra given in Figure 4. The medium range and relatively narrow bands with a transmittance of only about 15% and 40% seen at wave numbers of 2960 and 2876 cm$^{-1}$, which are attributed to C–H aliphatic symmetric and asymmetric stretching vibrations and the strong inplane bending vibrations with a transmittance of only about 5% at 1168 and 1064 cm$^{-1}$ wave numbers are attributed to the methyl groups [33,44]. The strong and broad peaks seen in the range of 2875–3600 cm$^{-1}$ are attributed to the quaternary amine salt formation with tetrafluoroborate (BF$_4^-$) of bmim[BF$_4$].

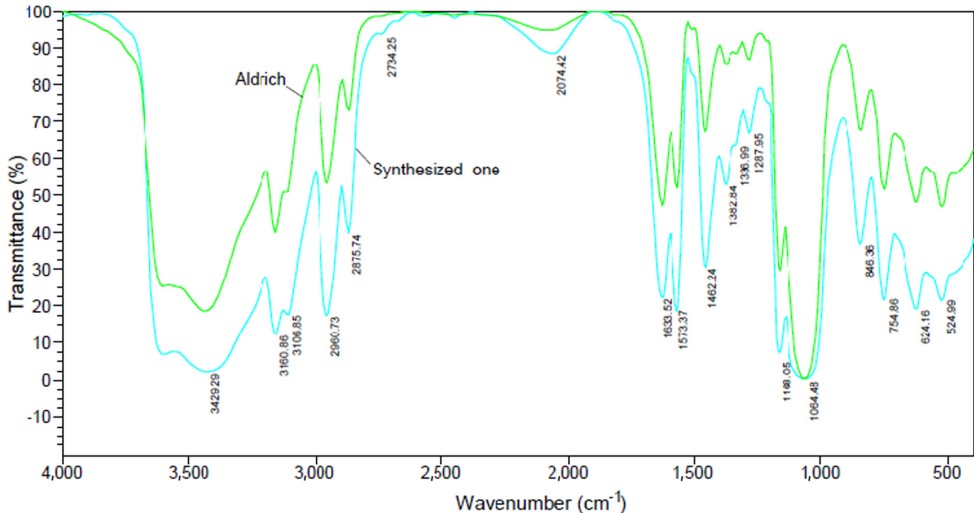

**Figure 4.** The FT-IR spectra (neat) of ARCI-bmim[BF$_4$] and Aldrich-bmim[BF$_4$].

The transmittance bands present between the wave numbers 1633 and 1462 cm$^{-1}$ are attributed to the C=N stretching of the imidazolium ring. Peaks appeared at wave numbers around 754 and 624 cm$^{-1}$ are attributed to the C–N stretching vibration [33,44]. The prominent FT-IR transmittance bands of various modes of sodium bromide (NaBr), which form as a byproduct during synthesis of bmim[BF$_4$], appear at wave numbers of 2957 cm$^{-1}$, 2946 cm$^{-1}$, 2928 cm$^{-1}$, 2855 cm$^{-1}$, 1624 cm$^{-1}$, 1617 cm$^{-1}$, 1464 cm$^{-1}$ and 1378 cm$^{-1}$ with a typical percentage transmittance of 5%, 6%, 4%, 9%, 34%, 32%, 25% and 38%, respectively [44]. None of the transmittance bands of Figure 4 match with those of NaBr with those kinds of percentage transmittance. Thus, the information gained from the FT-IR study is in line to the ones obtained from $^1$H- and $^{13}$C-NMR studies, which revealed that the in-house synthesized RTIL in this study was indeed bmim[BF$_4$]. Its mass spectral ($m^+/z$) study (Figure 5) also confirms that the synthesized ARCI-bmim[BF$_4$] RTIL is indeed bmim[BF$_4$] as the $m/z$ value of its bmim$^+$ [M + H]$^+$ moiety was recorded as 139.129 [33,44].

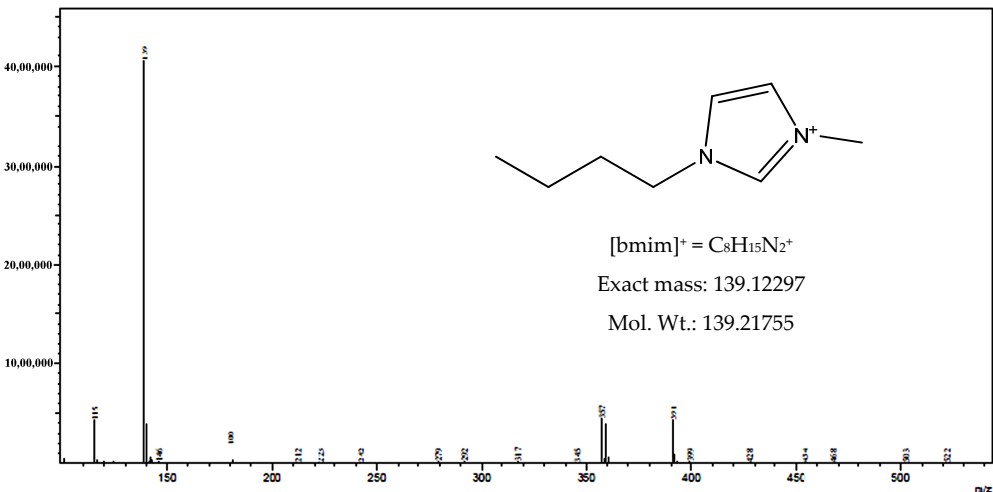

[bmim]$^+$ = C$_8$H$_{15}$N$_2$$^+$

Exact mass: 139.12297

Mol. Wt.: 139.21755

**Figure 5.** Mass spectrum of ARCI-bmim[BF$_4$] RTIL: HRMS (ESI): $m/z$ bmim$^+$ [M + H]$^+$: 139.

The ICP-OES studies revealed that the ARCI-bmim[BF$_4$] obtained in the continuous liquid–liquid extraction process contained about 32 ppm (i.e., 0.0032%) sodium (Na), whereas, the one obtained in a process without involving continuous liquid–liquid extraction step revealed the presence of about 46 ppm (i.e., 0.0046%) Na. On the other hand, the Aldrich-bmim[BF$_4$] exhibited the presence of about 20 ppm (i.e., 0.002%) Na.

### 3.2. Bmim[BF$_4$] Mediated ECR Reaction over MoSi$_2$ and Sn Cathodes

The CV recorded at 100 mV/s through 0 to −2.0 V vs. NHE for MoSi$_2$ WE dipped in the CO$_2$ saturated electrolyte solution made of either pure MeCN, MeCN + n-Bu$_4$NPF$_6$ (0.1 M) or MeCN + n-Bu$_4$NPF$_6$ + ARCI-bmim[BF$_4$] (20 mM) are presented in Figure 6a. It can be clearly seen from this figure that the higher current densities at considerably lower onset reduction potentials were generated over MoSi$_2$ WE when dipped in the CO$_2$ saturated MeCN electrolyte solution added with both n-Bu$_4$NPF$_6$ and bmim[BF$_4$] in comparison to those generated after dipping in either pure MeCN or in MeCN added with only the supporting electrolyte, n-Bu$_4$NPF$_6$ (0.1 M) but not the helper catalyst, bmim[BF$_4$]. These results indicate that the bmim[BF$_4$] helper catalyst is essential for achieving the current densities exceeding 100 mA/cm$^2$, which is a prerequisite for considering this process for commercial practice at industry. The CV recorded at the same 100 mV/s through 0.5 to −2.0 V vs. NHE for MoSi$_2$ WE (having a different exposed area to the electrolyte) after dipping in either Ar or CO$_2$ saturated electrolyte solution made of MeCN + n-Bu$_4$NPF$_6$ (0.1 M) + bmim[BF$_4$] (20 mM) are presented in Figure 6b. It can be seen from this figure that the higher current generation at lower onset reduction potentials was primarily due to the CO$_2$ reduction reaction as the amount of current generated for the electrolyte solution saturated with Ar is considerably lower than that of the one generated for the CO$_2$ saturated electrolyte. The current generated in the Ar saturated electrolyte has been reported to be due to the irreversible reduction of bmim[BF$_4$] or due to the hydrogen evolution reaction (HER) [2,4]. Since, in the present case, there is no water in the employed electrolyte solution, the generated current could be attributed mainly to the reduction of the bmim[BF$_4$] helper catalyst. As can be seen from Figure 6c, the current generation due to CO$_2$ reduction increased with the concentration of bmim[BF$_4$] in the catholyte solution. Furthermore, no change in the catalytic activity was also noted when the recycled bmim[BF$_4$] was employed for ECR reaction performed via CV over MoSi$_2$ WE (results not shown here).

The CV recorded at 100 mV/s through 0.5 to −2.0 V vs. NHE for Sn WE dipped in the electrolyte solution consists of MeCN and n-Bu$_4$NPF$_6$ (0.1 M) added with and without bmim[BF$_4$] (20 mM) after purging the electrolyte solution with either Ar or CO$_2$ gas indicate higher current densities at relatively lower onset reduction potentials for the electrolyte solution purged with CO$_2$ (red and green lines) in comparison to the one purged with the Ar gas (black and blue lines; Figure 7). It can also be seen that the presence of bmim[BF$_4$] had enhanced the generation of current densities. The peaks seen for electrolytes purged with CO$_2$ gas have been attributed to the CO$_2$ reduction reaction, whereas, those seen for electrolytes purged with Ar gas are attributed to the bmim[BF$_4$]'s irreversible reduction reaction and/or to the hydrogen evolution reaction (HER) as mentioned in the above-paragraph for MoSi$_2$ WE [1–5]. Rosenthal et al. [3] have also observed similar results for Sn WE nanoparticles deposited on a glassy carbon electrode (GCE) in CO$_2$ gas purged electrolyte solution of MeCN, n-Bu$_4$NPF$_6$ (0.1 M) and bmim[BF$_4$] (20 mM). It has been widely reported that for electrochemical reactions, which generate gaseous products, to be commercially successful for industrial practice, the products shall be formed at a rate of >100 mA/cm$^2$ current density [17,45–48]. The generation of about −110 mA/cm$^2$ (green line) at −1.052 V vs. NHE for CO$_2$ purged MeCN, n-Bu$_4$NPF$_6$ (0.1 M) and bmim[BF$_4$] (20 mM) was noted over Sn cathode (Figure 7 green line).

To confirm the reduction potential seen in the green line of Figure 6 and red line of Figure 7 was indeed due to the ECR reaction, the CPBE experiments were also conducted at −1.952 V vs. NHE in a two-compartment EC cell separated by nafion membrane by using the Sn or MoSi$_2$ bulk monolith cathode immersed in an catholyte solution made of MeCN, n-Bu$_4$NPF$_6$ (0.1 M) and ARCI-bmim[BF$_4$] (50 mM), which was purged and saturated with CO$_2$ gas and separated from an anolyte solution of 0.5 M NaPi (pH = 7.4) + 0.5 mM Co$^{2+}$ ions immersed with a Pt foil anode [46]. The expected reaction sequence that can occur in this CPBE experiment is schematically presented in Scheme 3 [1–5], and the FE, TOF and yield values of the products formed in these CPBE experiments are given in Table 3.

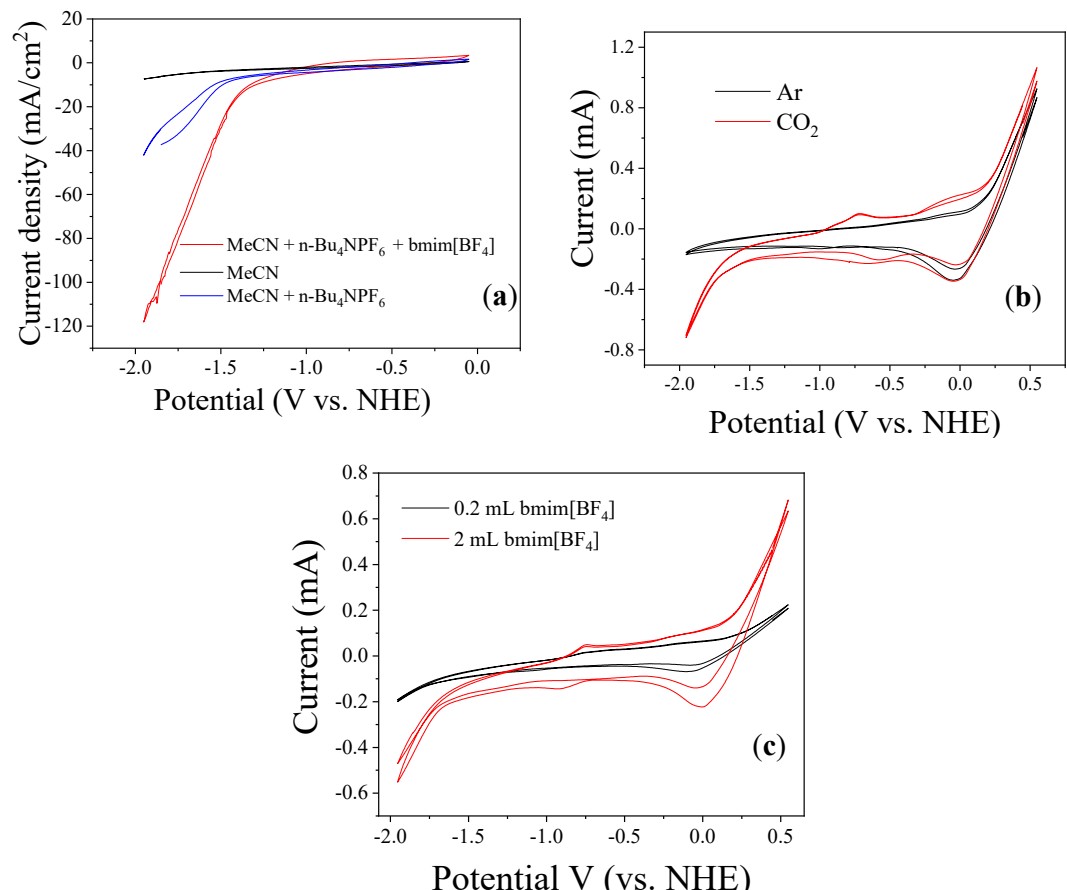

**Figure 6.** Cyclic voltammetry (CV) recorded at 100 mV/s over the MoSi$_2$ working electrode (WE) against the Pt wire counter electrode (CE) in (**a**) CO$_2$ purged MeCN, MeCN + n-Bu$_4$NPF$_6$ (0.1 M) and MeCN + n-Bu$_4$NPF$_6$ (0.1 M) + ARCI-bmim[BF$_4$] (20 mM) RTIL, (**b**) either CO$_2$ or Ar saturated electrolyte solution made of MeCN + n-Bu$_4$NPF$_6$ (0.1 M) + ARCI-bmim[BF$_4$] (20 mM) RTIL and (**c**) the CO$_2$ saturated electrolyte solution made of MeCN + n-Bu$_4$NPF$_6$ (0.1 M) + ARCI-bmim[BF$_4$] (0.2 or 2 mL).

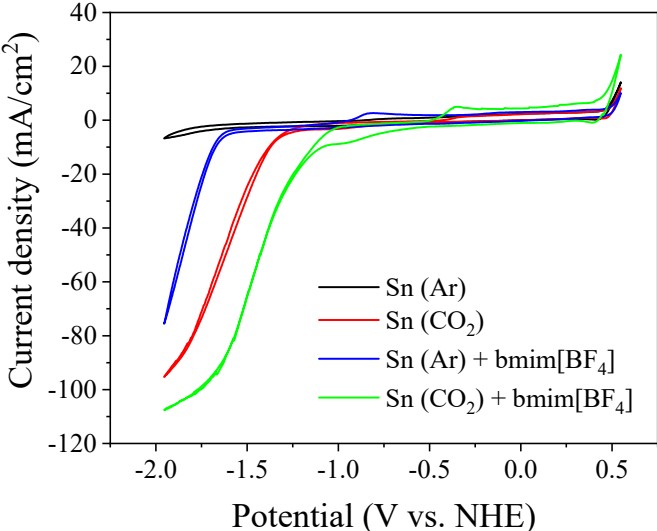

**Figure 7.** CV recorded at 100 mV/s over Sn WE against Pt wire CE in CO$_2$ or Ar purged MeCN + n-Bu$_4$NPF$_6$ (0.1 M) electrolyte added with and without ARCI-bmim[BF$_4$] (20 mM) RTIL.

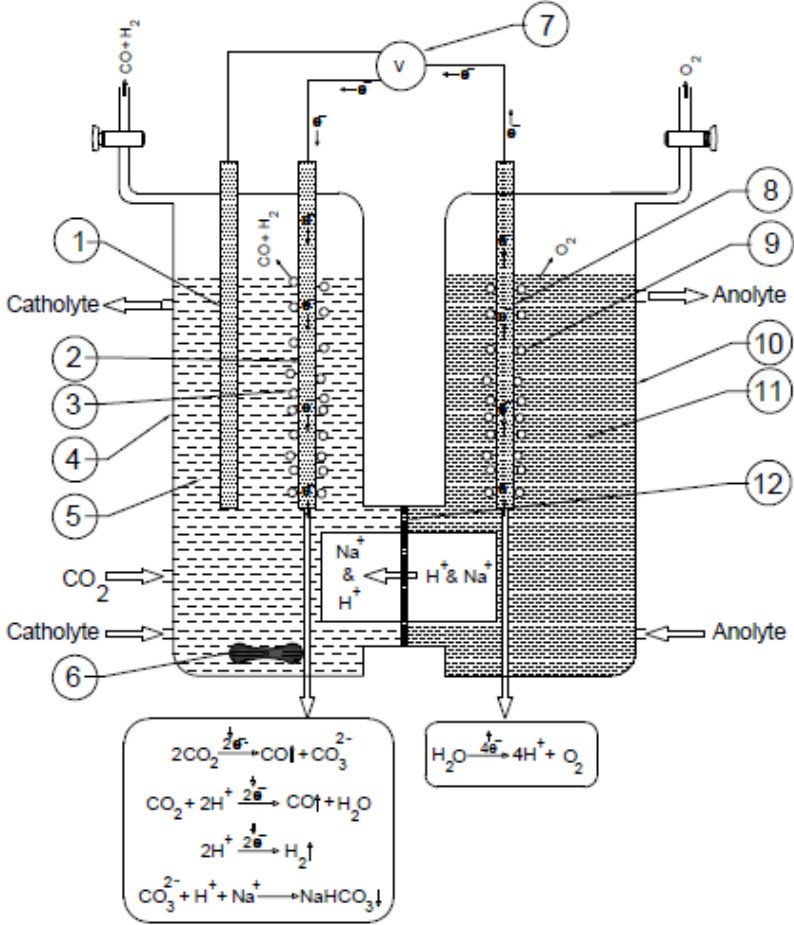

**Scheme 3.** A schematic diagram showing an electrochemical $CO_2$ reduction (ECR) reaction that can occur in a two-compartment electrochemical-cell on the surface of a metal cathode immersed in a $CO_2$ gas saturated and continuously purged aprotic non-aqueous catholyte solution (such as, MeCN + 0.1 M n-Bu$_4$NPF$_6$ + 50 mM bmim[BF$_4$]) separated by a proton exchange nafion membrane from an anolyte solution (such as, 0.5 M sodium phosphate (NaPi) buffer (pH = 7.4) containing 0.5 mM $Co^{2+}$ ions) to form CO and $H_2$ gases in a controlled potential bulk electrolysis (CPBE) experiment. (**1**) Reference electrode (Ag/Ag$^+$); (**2**) cathode (Sn); (**3**) CO and $H_2$ products; (**4**) cathodic compartment; (**5**) catholyte solution; (**6**) magnetic bead; (**7**) an electrochemical workstation; (**8**) anode (e.g., Pt); (**9**) $O_2$ gas; (**10**) anodic compartment; (**11**) anolyte solution and (**12**) nafion.

In a CPBE experiment performed as per the reaction conditions given in Table 3: 1st row (i.e., in the absence of $CO_2$ gas), the formation of $O_2$ and $H_2$ gases (Figure S7a-(A)) on the surfaces of the Pt anode and Sn cathode, respectively, was noted. In addition to $O_2$ evolution, a black/gray thin film formation on the surface of Pt anode was also noted [38,46]. This black/gray thin film has been identified to be a cobalt phosphate water oxidation catalyst (WOC), whose thickness was found to increase with the increase of CPBE time [38]. The presence of cobalt phosphate WOC as shown in Figure 8, whose SEM micrographs and energy dispersive analysis with X-rays (EDAX) spectrum are inserted in the current density and charge density profiles generated during its in situ formation on the surface of the Pt anode in the CPBE experiment conducted as per the reaction conditions given in Table 3: 1st row is very important to lower the overpotentials associated with water oxidation reaction (WOR) [38,46]. When the CPBE was performed in the presence of $CO_2$ gas (Table 3: 2nd row), the same $O_2$ evolution was observed in the headspace area of anodic compartment together with the formation of the black/gray colored thin film on the surface of anode, but in the headspace area of the cathodic compartment, instead of only $H_2$ gas, the formation of both CO (>95%) and $H_2$ (<5%) gases was noted (Figure S7a-(B)) [23,28]. The thermal conductivity response of $H_2$ gas was found to be almost

25 times higher than that of the one exhibited by CO gas (Figure S7b). The current density generation in this CPBE experiment was found to be strongly influenced by the concentration of $CO_2$ present in the catholyte solution [23,28]. When the catholyte solution was just saturated with $CO_2$ gas but not continuously purged during the entire CPBE experiment, a decrease in the current density values was noted after about 50 min of experiment, whereas, when the $CO_2$ gas was continuously purged during the entire CPBE experiment (Table 3: 3rd row), no decrease in the value of the current density until the completion of the experiment was noted [28]. A $FE_{CO}$ of 91.2%, yield of 75.10%, a TOF of about 0.75 s$^{-1}$ were noted when the CPBE experiment was conducted as per the reaction conditions given in Table 3: 3rd row for 180 min with a selectivity of 96% for CO and of 4% for $H_2$ formed over the surface of Sn bulk monolith cathode [28]. This observed TOF value is comparable with the one observed by Masel et al. [4]. The TOF values of 1–10 s$^{-1}$ are necessary for an ECR reaction to be practiced commercially [3]. No noticeable difference was observed in the results when ARCI-bmim[$BF_4$] was replaced with Aldrich-bmim[$BF_4$] CPBE (Table 3: 4th row).

**Table 3.** Results obtained in CPBE experiments carried out using Ar or $CO_2$ saturated and continuously purged catholyte solution of MeCN + 0.1 M n-Bu$_4$NPF$_6$ + 50 mM bmim[$BF_4$] over the Sn bulk-monolith cathode at an applied potential of −1.952 V vs. NHE that was separated from the Pt foil anode immersed in an anolyte solution of 0.5 M NaPi (pH = 7.4) buffer containing 0.5 mM $Co^{2+}$ ions by the nafion membrane [†].

| Entry | CPBE Time (min) | Current Density (Ma/Cm$^2$) | Charge Density (C/Cm$^2$) | FE (%; ±0.2) | | Selectivity (%; ±1) | | Mixture of CO + $H_2$ | Yield of CO (%) | Gases Formed (μMoles/Cm$^2$/H; ±1) | |
|---|---|---|---|---|---|---|---|---|---|---|---|
| | | | | $H_2$ | CO | $H_2$ | CO | TOF (1/s) | | $H_2$ | CO |
| 1. [‡] | 60 | 21.8 | 46.3 | 97.0 | 0 | >98 | 0 | 0.75 | 0 | 232.7 | 0 |
| 2. [¥] | 160 | 7.80 | 35.0 | 5.1 | 79.9 | 6 | 94 | 0.75 | 75.10 | 3.468 | 54.33 |
| 3. | 180 | 27.5 | 180.0 | 3.8 | 91.2 | 4 | 96 | 0.75 | 87.55 | 11.81 | 283.52 |
| 4. [§] | 160 | 16.8 | 258.0 | 4.62 | 87.87 | 5 | 95 | 0.75 | 83.47 | 23.18 | 440.50 |
| 5. [φ] | 160 | 14.7 | 39.5 | 6.67 | 76.63 | 8 | 92 | 0.75 | 70.49 | 3.667 | 51.37 |
| 6. [?] | 20 | 16.5 | 44.6 | 3.8 | 91.2 | 4 | 96 | 0.75 | 87.55 | 3.401 | 54.66 |

[†] The procedures employed to determine the values of FE, TOF, yield and selectivity as well as the amounts of gases formed are described in the supporting information (SI) file; [‡] the catholyte solution involved was saturated and continuously purged with high purity Ar gas; [¥] the catholyte solution involved was just saturated with $CO_2$ gas but not continuously purged during entire CPBE experiment; [§] involved Aldrich-bmim[$BF_4$] instead of ARCI-bmim[$BF_4$]; [φ] catholyte solution employed was a mixture of 90 vol.% MeCN + 0.1 M n-Bu$_4$NPF$_6$ + 50 mM bmim[$BF_4$] and 10 vol.% distilled water and [?] involved isotopic $^{13}CO_2$ gas as a reactant to form $^{13}CO$ gas.

The profile of the current generated over the $MoSi_2$ cathode during the CPBE experiment performed at an applied potential of −1.952 V vs. NHE in an all-glass two-compartment gas-tight electrochemical cell using a $CO_2$ saturated and continuously purged catholyte solution of MeCN + 0.1 M n-Bu$_4$NPF$_6$ + 50 mM ARCI-bmim[$BF_4$] separated from the Pt foil anode dipped in an anolyte solution of 0.5 M sodium phosphate (NaPi) buffer (pH = 7.4) containing 0.5 mM $Co^{2+}$ ions by a nafion membrane is given in Figure 9a as a function of the reaction time. Unlike the profiles generated over the Sn bulk monolith cathode under identical reaction conditions as reported elsewhere [28], for which the current generation was continuously increased until the completion of the CPBE period, whereas, in the present case, the current generated was almost constant over the surface of $MoSi_2$ cathode until the completion of the CPBE experiment performed for a period of about 40 min. A closer observation into these two separate results suggested that the current generation increases non-linearly with the reaction progress time until establishing the steady state conditions if the rate of $CO_2$ consumption is very high, which in turn is manifested by the continuous formation of a white precipitation in the catholyte solution with the progress of the CPBE period [28]. Recently, considerably high electrochemical $CO_2$ reduction activities were reported over various Mo based cathodic systems. In a study, Oh and Hu [30] have reported that $CO_2$ can be reduced to a mixture of products consisting of CO, formate and oxalate with varied concentrations depending on the reaction conditions employed over $MoO_2$/Pb in conjunction with 0.3M bmim[$BF_6$] at a temperature of −20 °C

or 21 °C with overpotentials as low as 40 mV. In another study, Asadi et al. [2] could reduce $CO_2$ to CO over $MoS_2$ insrted in a $CO_2$ saturated catholyte solution made of 4 mole% aqueous bmim[$BF_4$] RTIL at a overpotential of only 54 mV with a FE of >95%, and current density of 40 mA/cm$^2$. Yet, in another study, Handoko et al. [31] have reported that $Mo_2CT_x$ MXenes can activate $CO_2$ and reduce it to form formic acid with a selectivity of about 32.6% at a rate formation of 2.5 mA/cm$^2$ current density at 1.3 V vs. SHE. The different catalytic activities observed over Mo, $MoO_2$, $MoS_2$, $Mo_2CT_x$ MXenes, etc., in comparison to $MoSi_2$ can be attributed to the presence of different kinds of catalytically active sites on these molybdenum based cathodic systems. These results suggest that similar to the Mo metal cathode, the $MoSi_2$ can also be employed as a physically robust cathodic system in conjunction with bmim[$BF_4$] RTIL for the ECR to CO formation reaction with reasonably high catalytic activity.

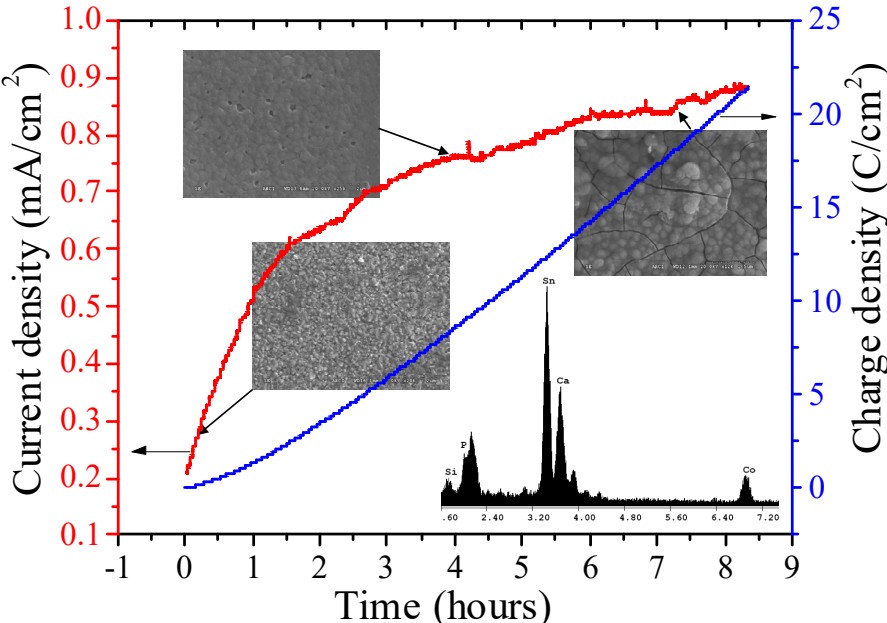

**Figure 8.** The profiles of the current density versus time and charge density versus time generated during a CPBE experiment performed as per the reaction conditions given in the 1st row of Table 3. The accompanying SEM micrographs and energy dispersive analysis with X-rays (EDAX) spectra belong to the in situ formed gray colored cobalt phosphate (CP) thin film on the surface of the Pt anode immersed in an anolyte solution of 0.5 M sodium phosphate (NaPi) buffer (pH = 7.4) containing 0.5 mM $Co^{2+}$ ions, and were recorded after completing the experiment for different time intervals.

The Tafel slopes generated for CV measured over WEs made of Sn (as shown in Figure 7 by a green colored line), and of $MoSi_2$ (as shown in Figure 6a by a red colored line) dipped in $CO_2$ saturated and blanketed solution of MeCN + 0.1 M n-Bu4NPF$_6$ + 20 mM bmim[$BF_4$] against Pt wire CE at a scanning rate of 100 mV/s according to a relation given in Equation S9 are presented in Figure 9b. A closer look into these Tafel slope values indicate that both are higher than the value of 118 mV/decade, suggesting that the one-electron transfer reaction is not the rate-determining step for this reaction. The overpotentials associated with the Sn cathode (as can be seen from Figure 7) were much lower than the one exhibited by $MoSi_2$ cathode as can be seen from Figure 6a. The higher Tafel slope value (360 mV/decade) noted for $MoSi_2$ cathode suggests that there is a large resistance for exchanging of electrons (e$^-$) between the electrode surface and the reactive species present in the electrolyte solution [28].

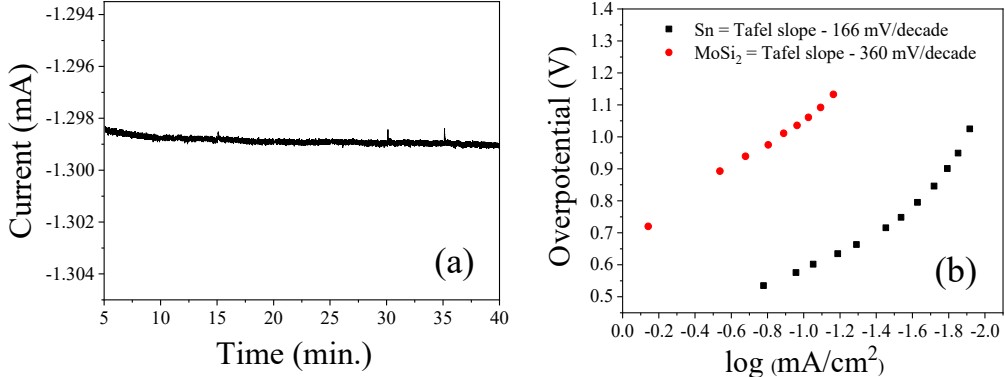

**Figure 9.** (**a**) A profile of current generated vs. time during the CPBE experiment conducted using MoSi$_2$ as a cathode immersed in a CO$_2$ saturated catholyte solution of MeCN + 0.1 M n-Bu$_4$NPF$_6$ + 50 mM ARCI-bmim[BF$_4$] separated from a Pt foil anode dipped in an anolyte solution of 0.5 M sodium phosphate (NaPi) buffer (pH = 7.4) containing with 0.5 mM Co$^{2+}$ ions by a nafion membrane at an applied potential of −1.952 V vs. NHE; (**b**) Tafel slopes generated for CV measured over WEs made of Sn as shown in Figure 7 (green colored line) and over MoSi$_2$ as shown in Figure 6a (red colored line) dipped in CO$_2$ saturated and blanketed solution of MeCN + 0.1 M n-Bu4NPF$_6$ + 20 mM bmim[BF$_4$] against Pt wire CE at a scanning rate of 100 mV/s according to a relation given in Equation (S9) (given in the associated supporting information file).

The SEM micrograph of the centre cut-portion of the MoSi$_2$ heating element employed as the cathode to perform the ECR to CO formation reaction, and its corresponding EDAX spectrum are given in Figure 10a,b, respectively. The EDAX spectrum indicates that this sample was indeed the MoSi$_2$ and no other elemental signals as impurities can be seen from this spectrum. The SEM micrograph shown in Figure 10a matched very well with those reported for MoSi$_2$ sintered body [32]. A very small fraction of porosity (<3%) and the presence of no glassy phase near grain boundaries suggest that the employed MoSi$_2$ was a high-purity and high-density body. From this micrograph, the presence of some pores formed due to the grain pull mechanism can also be seen. Nevertheless, when ECR reaction activity exhibited by Sn and MoSi$_2$ cathodes were compared, although both of them exhibited current densities exceeding >100 mA/cm$^2$ during CV measurements, the overpotential associated with Sn was much lower than the one noted for MoSi$_2$.

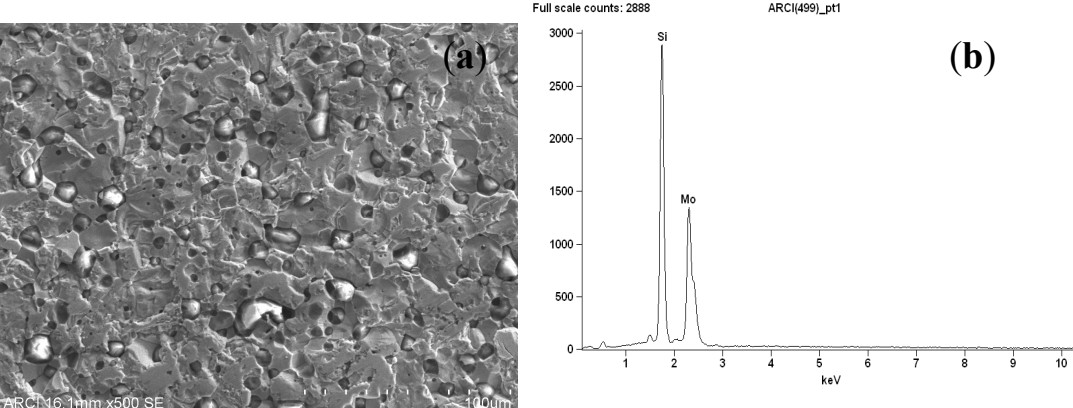

**Figure 10.** The SEM micrograph (**a**) and the EDAX spectrum (**b**) of the dense MoSi$_2$ surface of a centre cut portion of 8 mm (φ) diameter used Kanthal® Super 1900 Heating Element.

Surprisingly, the in situ formation of a white solid substrate along with CO and H$_2$ gases on the surface of the Sn cathode during CPBE of bmim[BF$_4$] mediated ECR was also noted, whose concentration in the cathodic compartment was continuously increased with the increase of the CPBE time. When the CPBE was performed for a period of 180 min as per the reaction conditions

given in Table 3: 3rd row, the complete catholyte solution was filled with white precipitate, which was characterized to be a pure compound of $NaHCO_3$ by XRD analysis (Figure S8) [45]. The formation of CO together with formic acid (HCOOH) on the interface of Sn/SnO at −0.7 V vs. RHE was reported by Chen and Kanan [48] when the $CO_2$ saturated aqueous $NaHCO_3$ catholyte was separated from the same aqueous $NaHCO_3$ anolyte solution by a nafion membrane upon the electrochemical reduction.

To determine the influence of water on the product formation in the ECR reaction over the Sn cathode, another CPBE experiment was also performed as per Table 3: 5th row reaction conditions, in which about 10 vol.% of the catholyte solution made of MeCN, n-$Bu_4NPF_6$ (0.1 M) and bmim[$BF_4$] (50 mM) was replaced with pure distilled water, and the white precipitate formed in the cathodic compartment was analyzed by XRD and FT-IR techniques. For comparison purposes, the XRD pattern and FT-IR spectrum of the white compound in situ formed in CPBE as per Table 3: 5th row reaction conditions were compared with those of HCOONa formed out of NaOH and HCOOH in Figures 11 and 12, respectively, together with those recorded for the white compound formed in the catholyte solution containing no water (i.e., in the CPBE experiment conducted as per the conditions given in Table 3: 3rd row).

A close observation at XRD patterns (Figure 11) and FT-IR spectra (Figure 12) reveals that the presence of water in the catholyte solution promotes the formation of formic acid (i.e., in this case it is sodium formate) as well along with the formation of $NaHCO_3$ on the surface of Sn cathode in the bmim[$BF_4$] mediated ECR reaction [48]. Hass et al. [45] have reported that the cations such as, $Na^+$, $Co^{2+}$, etc., present in the anolyte solution are transported via the nafion membrane during the CPBE experiment until all of them are completely transported from the anolyte to catholyte solution leaving only pure water as the anolyte solution.

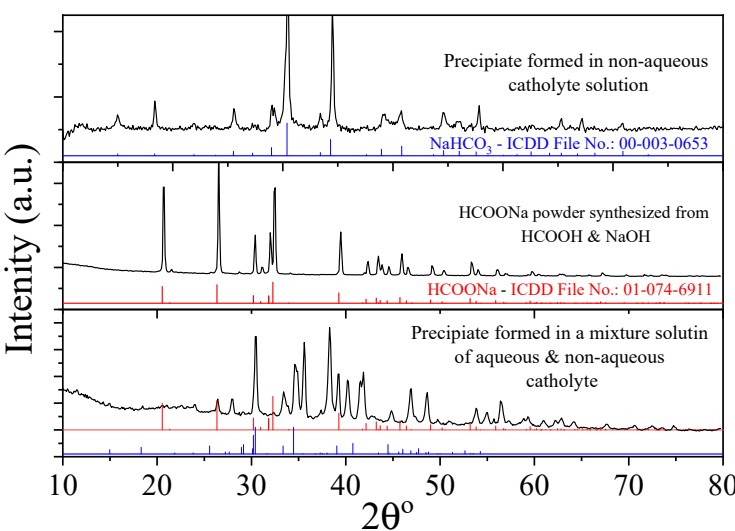

**Figure 11.** XRD patterns of white precipitates in situ formed in the cathodic compartment during CPBE experiments performed as per reaction conditions given in the 3rd and 5th rows of Table 3 along with the one of HCOONa synthesized from HCOOH and NaOH.

Owing to its planar $XY_3$ structure, $CO_3^{2-}$ ion exhibited $D_{3h}$ point group symmetry, its $\nu_1$, $\nu_3$ and $\nu_4$ modes were Raman active and $\nu_2$, $\nu_3$ and $\nu_4$ were IR active [49]. Due to the presence of carbonyl (−C=O) and hydroxyl (−OH) groups, $NaHCO_3$ exhibited a slightly broad and medium range IR transmission band between 1680 and 1750 $cm^{-1}$, and another medium range broad IR band between 3230 and 3550 $cm^{-1}$, respectively [49]. The formic acid owing to its C−H and O−H bonds exhibited IR bands in the range of 2850–3300 $cm^{-1}$, and 2500–3000 $cm^{-1}$, respectively. The white precipitate formed in the CPBE reaction as per Table 3: 3rd row reaction conditions exhibited an IR transmission band around 1720 $cm^{-1}$ and another wide band with a medium height between 3000 and 3500 $cm^{-1}$, which indicate the presence of the $NaHCO_3$ compound (Figure 12). On the other hand, the white

precipitate formed in the water-containing catholyte solution (Table 3: 5th row reaction conditions) exhibited IR bands due to the presence of both $NaHCO_3$ and $HCOOH$ compounds. These results suggest that the non-aqueous aprotic catholyte solution of MeCN, n-$Bu_4NPF_6$ (0.1 M) and bmim[$BF_4$] (50 mM) do not support the formation of any other product including $HCOOH$ except for the formation of CO and $H_2$ gases together with $NaHCO_3$ when the 0.5 M NaPi (pH = 7.4) buffer containing 0.5 mM $Co^{2+}$ ions was employed as the anolyte solution and was separated by a nafion membrane upon reduction over the Sn cathode at −1.952 V vs. NHE [11,50].

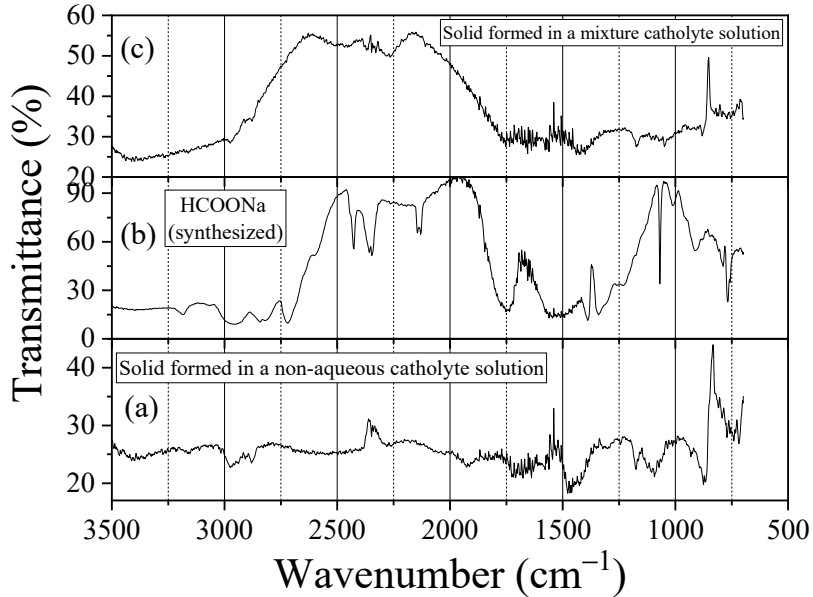

**Figure 12.** FT-IR spectra of white precipitates in situ formed in the cathodic compartment during CPBE experiments performed as per the reaction conditions given in the 3rd and 5th rows of Table 3 along with the one of HCOONa synthesized from HCOOH and NaOH.

The $^1$H-NMR spectral investigation (spectrum given two different scales in Figure S9a,b of catholyte solution of the CPBE experiment performed as per Table 3: 3rd row reaction conditions after separating the solid substance from it by a filtration process and rota-evaporation) indicates the presence of no other compounds other than MeCN, n-$Bu_4NPF_6$, and bmim[$BF_4$], which are the components of catholyte solution. The $HCOOH$ and HCCONa compounds exhibited an NMR singlet signal at a δ value of 9.6 ppm, whereas, Figure S9b revealed no NMR signals at the δ value of 9.6 ppm. These results indicate that no liquid products are formed in bmim[$BF_4$] mediated ECR reaction, when it is performed over the Sn cathode under reaction conditions given in Table 3: 3rd row.

The SEM of the fresh surface of Sn cathode and after it was employed as a cathode to perform the CPBE reaction as per the Table 3: 3rd row reaction conditions are presented in Figure 13a,b, respectively, whose XRD patterns are given in Figure 13c. A closer look into the morphological features of the Sn cathode before and after employing it as a cathode in the ECR reaction indicates that the Sn cathode was physically and morphologically quite stable and did not undergo any noticeable changes after $CO_2$ underwent reduction to CO gas or after $H_2$ and $NaHCO_3$ formation occurred on its surface [3,51–54]. XRD patterns of Sn also revealed the same information that is obtained from SEM analysis.

To confirm whether CO was formed out of only $CO_2$ gas or from any other organic matter that was present in the reaction medium, a separate CPBE experiment was also conducted as per the conditions given in Table 3: 6th row, in which an isotopic $^{13}C$-$CO_2$ gas was used as a reactant. The mass spectra of CO gases formed in the CPBE experiments performed as per the reaction conditions given in 6th and 3rd rows of Table 3 are given in Figure 14a,b, respectively. The presence of the little fraction of $^{12}C$-CO gas ($m/z^+$ = 28.1; Figure 14a) in $^{13}C$-$CO_2$ gas involved CPBE experiment can be seen with the

rest being only due to $^{13}$C-CO gas ($m/z^+ = 29.1$). Thus, the ECR experiments conducted in this study using isotopic $^{13}$C-CO$_2$ gas confirmed that CO was the product of only CO$_2$ gas and not the product of any other organic matter present in the system. Figure 15 shows the digital photographs of various electrochemical cells and apparatus employed to perform ECR reactions in this study including those with isotopic $^{13}$C-CO$_2$ gas (the $^{13}$C-CO$_2$ isotopic gas cylinder can be seen from Figure 15d.

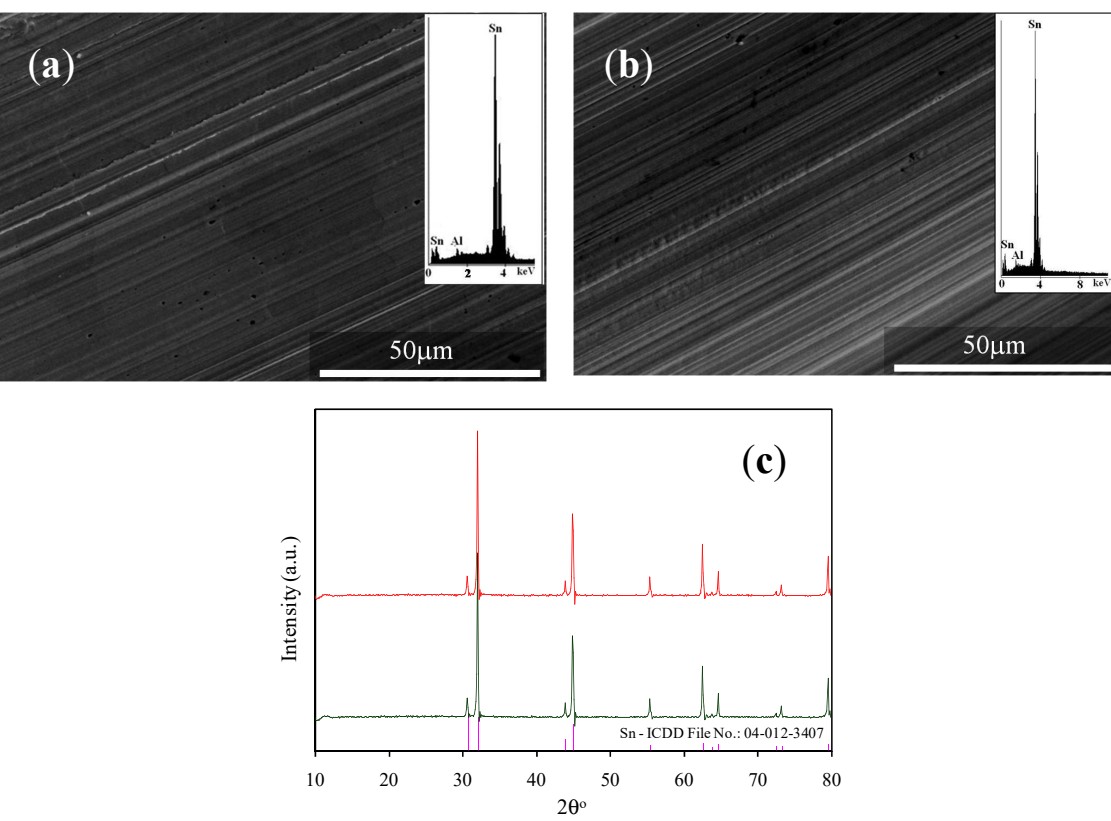

**Figure 13.** SEM micrographs and EDAX spectra of Sn metal before (**a**) and after being (**b**) employed as a cathode in the CPBE experiment performed as per the reaction conditions given in the 3rd row of Table 3, and (**c**) the XRD patterns of Sn before and after it was employed as a cathode in the CPBE experiment performed as per the reaction conditions given in the 3rd row of Table 3.

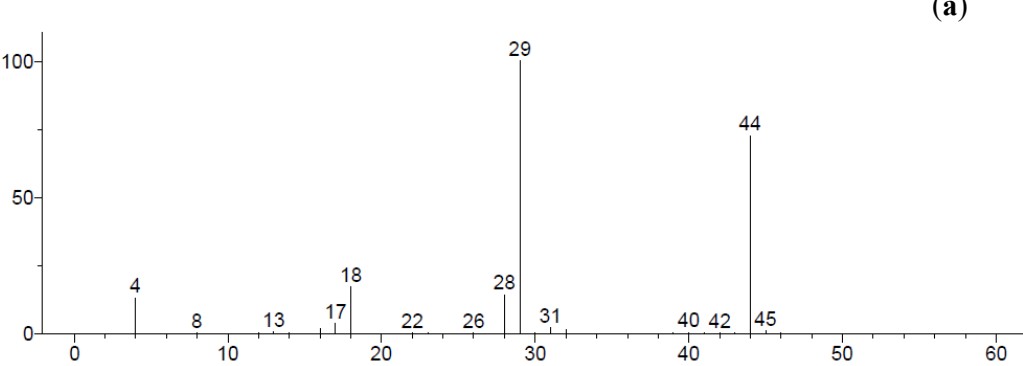

**Figure 14.** *Cont*.

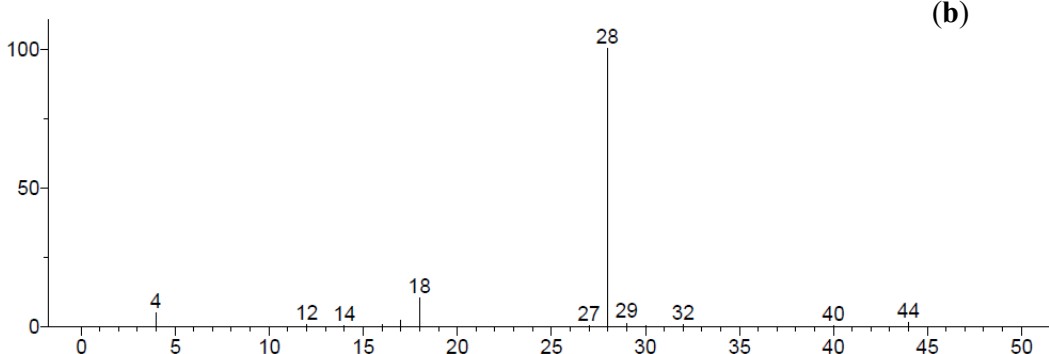

**Figure 14.** Mass spectra of $^{13}C$-CO (**a**) and $^{12}C$-CO (**b**) gases formed over the Sn cathode surface during ECR reactions performed in CPBE experiments as per the reaction conditions given in the 6th and 3rd rows of Table 3, respectively.

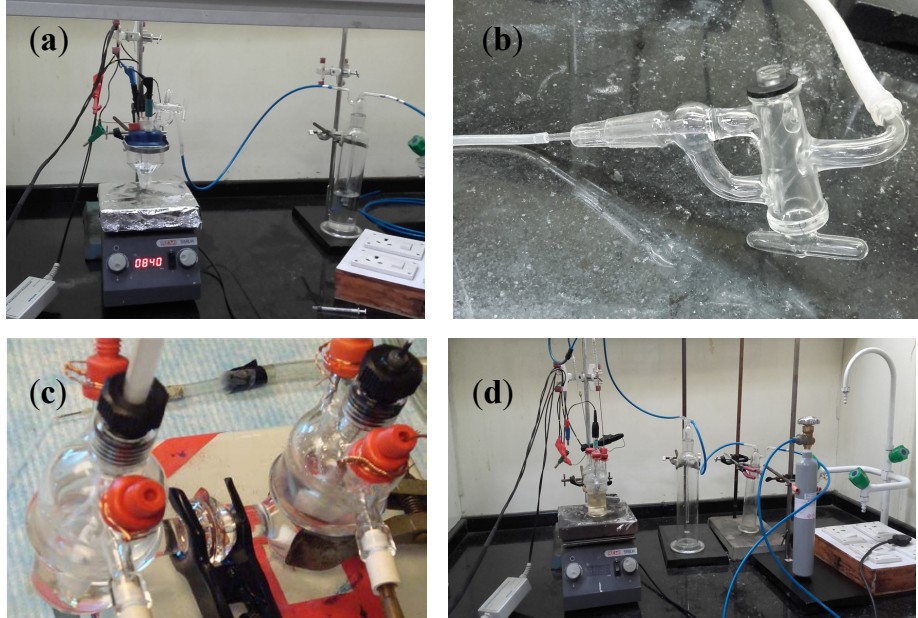

**Figure 15.** Digital photograph showing (**a**) a set-up employed for conducting cyclic voltammetry (CV) experiments, (**b**) a 3-way glass stop-cock used for gas purging and blanketing of electrolyte solution during CV measurements, (**c**) a two-compartment electrochemical (EC) cell separated by the nafion membrane employed for performing the ECR reaction in the CPBE process and (**d**) a set-up employed for performing CPBE of $^{13}CO_2$ gas in a two-compartment EC cell.

The catholyte solution pH change information can also be used to assess whether the $NaHCO_3$ compound can form or not on the surface of the Sn cathode in the bmim[$BF_4$] ECR reaction when performed via CPBE according to the reaction conditions given in Table 3: 2nd to 5th rows. When the CPBE experiment was performed as per Table 3: 5th row conditions, the initial pH of the catholyte solution made of 65 mL MeCN + 0.1M n-$Bu_4NPF_6$ + 50 mM bmim[$BF_4$] and 10 mL distilled water was about 2.02, whose pH was increased to 7.8 after 3 h of CPBE, and to 8.38 after another additional 3 h. Whereas, the pH of the MeCN was 5.8, and the pH of MeCN + 0.1 M n-$Bu_4NPF_6$ + 50 mM bmim[$BF_4$] was about 3.39, whose pH was raised to about 6.82 after performing CPBE experiment for 6 h according to Table 3: 3rd row conditions. This pH rise of catholyte solution suggests that a basic compound, in this study, $NaHCO_3$ as confirmed by the XRD study, is in situ formed in the bmim[$BF_4$] mediated

ECR reaction when performed over the Sn cathode after dipping it in a $CO_2$ saturated and purged non-aqueous aprotic solvent based catholyte solution that was separated from the aqueous 0.5 M NaPi (pH = 7.4) buffer solution containing 0.5 mM $Co^{2+}$ ions by nafion membrane at −1.952 V vs. NHE.

The in situ formed $H_2O$ and $NaHCO_3$ can dissolve together in the catholyte solution, and cause the rise of solution pH as both $H_2O$ and MeCN were highly miscible. The particles of $NaHCO_3$ settled down at the bottom of the catholyte solution in the cathodic compound when its particle size grew to the extent where they were pulled down by the gravitational force.

The $KHCO_3$ formation during an ECR reaction performed in an aqueous $KHCO_3$ solution employed as both an anolyte and catholyte and separated by a nafion (i.e., proton exchange) membrane was also noted by Haas et al., [45]. It was reported that in this aqueous $KHCO_3$ electrolyte, the reaction proceeded as per the sequence given in Equations (12)–(14), whereas, the reaction given in Equation (14) was most unlikely to take place as $2HCO_3^-{}_{(aq)}$ did not pass through the proton exchange nafion membrane to undergo the oxidative splitting to form $CO_2$ and $H_2O$ in combination with protons ($H^+$) present over there.

$$\text{Cathodic reaction: } 3CO_2 + H_2O + 2e^- \rightarrow CO\uparrow + 2HCO_3^-{}_{(aq)} \tag{12}$$

$$\text{Anodic reaction: } H_2O - 2e^- \rightarrow \frac{1}{2}O_2\uparrow + 2H^+ \tag{13}$$

$$\text{Anodic reaction: } 2HCO_3^-{}_{(aq)} + 2H^+ \rightarrow 2CO_2\uparrow + H_2O \tag{14}$$

The results obtained in the CV measurements in water-free $CO_2$ purged electrolyte formed out of MeCN, n-$Bu_4NPF_6$ (0.1 M) and bmim[$BF_4$] (20 mM) over Sn WE as presented in Figure 7 as the green line, and the results obtained in CPBE experiments as summarized in Table 3 that include (i) $H_2$ (5%; as per Equation (15)) + CO (95%) gases formation, (ii) $NaHCO_3$ and CO formation in almost 1:1 ration, (iii) transportation of $Co^{2+}$ and $Na^+$ ions along with protons ($H^+$) through nafion membrane from the anolyte to catholyte solution, (iv) low hydration equilibrium constant ($K_r = 1.7 \times 10^{-3}$) for $CO_2$ + $H_2O \rightarrow H_2CO_3$ reaction at 25 °C (Equation (1)) and (v) the acid dissociation constants of $H_2CO_3$ (Equations (2) and (3)) suggest that protons ($H^+$) formed in the WOR on the surface of the Pt anode in the anodic compartment cannot participate in the bmim[$BF_4$] mediated ECR to CO formation reaction over the Sn cathode as per the reaction given in Equation (16). This is due to the fact that if the ECR reaction proceeds as per Equation (16), then there cannot be any formation of $H_2$ gas as all the electrons ($e^-$) generated from water in the WOR (Equation (17)) are consumed in the CO formation itself (Equation (16)) and no electrons ($e^-$) shall be available to react with protons ($H^+$) to form $H_2$ gas as per Equation (15), whereas, the formation up to 5% $H_2$ was noted when CPBE experiments were conducted as per the conditions given in Table 3: 2nd to 5th rows. This is due to the fact that if ECR reaction proceeds as per Equation (16), two-electrons ($e^-$) are required to form each CO molecule, whereas, in corresponding and simultaneously and concurrently occurring WOR on the surface of the Pt anode in the anodic compartment, two-electrons ($e^-$) are generated along with two-protons ($H^+$) as per Equation (15). In fact, all the electrons ($e^-$) that travel from anode to cathode via an external circuit cannot receive an equivalent number of protons ($H^+$) from the anodic compartment to the cathodic compartment via the nafion membrane, as there is competition from $Co^{2+}$ and $Na^+$ cations as well as to the transportation of protons ($H^+$). These results suggest that the CO formation in this ECR reaction can be assumed to be an electrochemical reverse reaction of CO oxidation in air to form $CO_2$ gas that does not involve any protons ($H^+$) in the oxidation process.

$$2H^+ + 2e^- \rightarrow H_2\uparrow; E^0 = 0 \text{ V vs. NHE} \tag{15}$$

$$CO_2 + 2H^+ + 2e^- \rightarrow CO\uparrow + H_2O; E^0 = -0.10 \text{ V vs. NHE} \tag{16}$$

$$2H_2O \rightarrow 4H^+ + 4e^- + O_2\uparrow; E^0 = -1.23 + 0.059 \text{ (pH) V vs. NHE} \tag{17}$$

The $CO_2$ reduction activity noted in CV (Figure 7 green line) measurements performed in water-free non-aqueous aprotic catholyte solution also suggests that $CO_2$ can undergo reduction in the absence of protons ($H^+$). As up to 5% $H_2$ formation was noted in CPBE experiments as shown in Table 3, the ECR might not occur as per the reaction given in Equation (16). In addition to that the prevalent reaction conditions during CPBE experiments performed as per the reaction conditions given in Table 3 do not allow the formation of $NaHCO_3$ according to the reactions given in Equations (2)–(4) (refer to the introduction part) as the hydration equilibrium constant ($K_r$) was very low ($1.7 \times 10^{-3}$ (Equation (1)), at which, only a small fraction of $CO_2$ gas pumped into the catholyte solution could react with $H_2O$ (if water is in situ formed as per the reaction given in Equation (16)) to form $H_2CO_3$ at a very low reaction rate (slow kinetics).

Actually, the majority of the $CO_2$ gas pumped into the catholyte solution was not converted into $H_2CO_3$, and it will remain as $CO_2$ gas only. In the absence of any catalyst, the $CO_2$ hydration equilibrium is reached quite slowly. Furthermore, the rate constants for $CO_2 + H_2O \rightarrow H_2CO_3$ (forward reaction) and $H_2CO_3 \rightarrow CO_2 + H_2O$ (reverse reaction) were 0.039 $s^{-1}$ and 23 $s^{-1}$, respectively. Apart from these, the $pK_{a1}$ is 6.363 (i.e., at a pH of 6.363, only 50% of $H_2CO_3$ formed in the catholyte solution is dissociated into $HCO_3^-$ ions and protons ($H^+$)). In such a case, the lower number of $HCO_3^-$ and $Na^+$ ions present in the catholyte solution could not lead to the formation of $NaHCO_3$ in a 1:1 ration with CO gas formed. Furthermore, in these reaction conditions, $CO_3^{2-}$ could not form as $pK_{a2}$ is about 10.329 as shown in Equation (3) (introduction part).

In view of the above mentioned reasons, the bmim[$BF_4$] mediated ECR to CO formation reaction over the Sn cathode in non-aqueous aprotic catholyte solution can be considered as a reverse reaction of CO oxidation in air as it can occur as per the reaction sequence as shown in Equations (18)–(20), and this reaction cannot proceed as per the reaction mechanism proposed by Wang et al. [54] (Scheme S1) but as per the one given in Schemes 4 and 5 [55–60].

$$2CO_2 + 2e^- \rightarrow CO\uparrow + CO_3^{2-}; E^0 = -0.449 \text{ V vs. NHE} \tag{18}$$

$$CO_3^{2-} \rightarrow CO_2 + \frac{1}{2} O_2\uparrow + 2e^-; E^0 = -0.883 \text{ V vs. NHE} \tag{19}$$

$$\text{The net reaction: } CO_2 \rightarrow CO + \frac{1}{2} O_2; E^0 = -1.33 \text{ V vs. NHE} \tag{20}$$

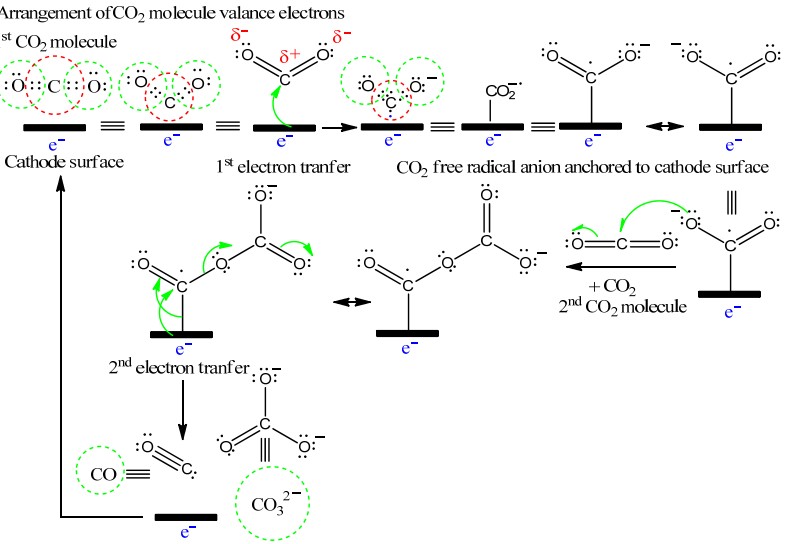

**Scheme 4.** A schematic diagram depicting a plausible reaction mechanism proposed that is believed to takes place in an ECR to CO formation reaction in a non-aqueous aprotic-solvent based catholyte solution on the surface of an inert metal cathode such as Sn and $MoSi_2$.

**Scheme 5.** A schematic diagram showing a plausible reaction mechanism to be involved in an imidazolium based RTIL mediated electrochemical $CO_2$ reduction (ECR) to form CO in a non-aqueous aprotic-solvent based catholyte on the surface of the Sn cathode (for clarity purposes, the $BF_4^-$ anion is deleted from the structures).

Medina-Ramos et al. [3] and Chen and Kanan [48] have explained the role of Sn on efficient reduction of $CO_2$ to CO. It has been identified that the interface of Sn/SnO thin layers are the actual catalytically active sites that activate the $CO_2$ molecule [48]. It has been established that in the absence of the desired reducing atmosphere, at potentials higher than those of the standard reduction potential (SRP), still some oxide groups are present on the surface of metal cathodes [59]. Such oxide layers present on the surface of the Sn cathode could be responsible for the noted high activity for the bmim[$BF_4$] mediated ECR to CO formation reaction performed in an aprotic non-aqueous catholyte solution.

## 4. Conclusions

An high-purity bmim[$BF_4$] RTIL was synthesized by following a simple and inexpensive route, and thoroughly characterized by means of $^1$H- and $^{13}$C-NMR, FT-IR, mass and ICP-OES techniques. The electrocatalytic activity of synthesized bmim[$BF_4$] was established vis-a-vis Aldrich-bmim[$BF_4$] for ECR to CO formation over Sn and MoSi$_2$ bulk monolith cathodes. No noticeable difference was noted for ECR to CO formation activity when the ARCI-bmim[$BF_4$] was replaced by Aldrich-bmim[$BF_4$]. When the $CO_2$ purged aprotic non-aqueous catholyte solution of MeCN, n-Bu$_4$NPF$_6$ (0.1 M) and bmim[$BF_4$] (50 mM) was separated by a nafion membrane from the anolyte solution of 0.5 M sodium phosphate (NaPi) buffer (pH = 7.4) containing with 0.5 mM $Co^{2+}$ ions, and subjected to a reduction process at an applied potential of −1.952 V vs. NHE (i.e., at 622 mV overpotential) against the Pt foil anode in a CPBE experiment, the formation of CO with a selectivity of 96%, current density of 27 mA/cm$^2$, faradaic efficiency of 91.2% and yield of 87.55% was noted. The bmim[$BF_4$] mediated ECR to CO formation has been identified to be an electrochemical reverse reaction of CO oxidation in air that does not involve protons (H$^+$) in the reduction process. Isotopic $^{13}$C-$CO_2$ gas involved CPBE experiments revealed that CO is a product of only $CO_2$ gas and not from any of the other organic matter present in the reaction mixture. The catalytic activity of the MoSi$_2$ cathode was found to increase with the increased concentration of the bmim[$BF_4$] helper catalyst in the catholyte solution under a given applied reduction potential. Similar to the fresh bmim[$BF_4$] RTIL, the recycled one also exhibited the same activity for the ECR to CO formation reaction over the MoSi$_2$ cathode under identical reaction conditions. The onset reduction potential noted for the ECR reaction over the Sn cathode is considerably lower than the one revealed by MoSi$_2$ under identical reaction conditions.

**Supplementary Materials:** The following are available online at http://www.mdpi.com/2311-5629/6/3/47/s1, Figure S1: XRD pattern of NaBr formed as a byproduct during synthesis of bmim[BF$_4$] RTIL from bmim[Br] and NaBF$_4$ via metathesis reaction performed at room temperature in water medium along with ICDD file # 00-005-0591 assigned for NaBr compound, Figure S2: A digital photograph showing the continuous liquid-liquid extraction apparatus employed for separating bmim[BF$_4$] RTIL from NaBr salt from the reaction mixture solution. Figure S3: CV profile generated while calibrating Ag/Ag$^I$ (0.01M) reference electrode in Ar saturated and blanketed solution of 20 mL MeCN + 200 mM n-Bu$_4$NPF$_6$ + 5 mM Ferrocene at a scan rate of 20 mV/s over Pt working (WE) and counter electrodes (CE). Figure S4: (a) $^1$H-NMR spectrum of bmim[Br] RTIL synthesized in this study (refer Table 2), & (b) $^{13}$C-NMR (125 MHz, CDCl$_3$ + DMSO) spectrum of bmim[Br] compound synthesized in this study (refer Table 2). Figure S5: (a) $^1$H-NMR of bmim[BF$_4$] RTIL prepared in this study (refer Table 2), & (b) $^{13}$C-NMR (125 MHz, CDCl$_3$ + DMSO) of bmim[BF$_4$] RTIL prepared in this study (refer Table 2). Figure S6: (a) $^1$H NMR (400 MHz, CDCl$_3$) spectrum of bmim[BF$_4$] ($\geq$97.0%, 91508-5G) RTIL procured from Aldrich, USA (refer Table 2), & (b) $^{13}$C-NMR spectrum of bmim[BF$_4$] ($\geq$97.0%, 91508-5G) RTIL procured from Aldrich, USA (refer Table 2). Figure S7: (a) Gas chromatograph (GC) profiles gases formed in cathodic compartment during a CPBE experiment performed as per reaction conditions given 1st and 3rd rows of Table 3, respectively; and (b) the response factors of H$_2$ and CO gases by thermal conductivity detector (TCD). Figure S8: XRD pattern of a white precipitate in situ formed in the cathodic compartment during CPBE experiment conducted as per the reaction conditions given in the 3rd row of Table 3. All the major XRD peaks exhibited by this white precipitate are matching with those lines of NaHCO$_3$ published in ICDD File # 00-003-0653. Figure S9: (a) $^1$H NMR (400 MHz, CDCl$_3$) spectrum of concentrated catholyte solution of CPBE experiment performed as per the reaction conditions given in 3rd row of Table 3: $\delta$ (ppm) = For bmim[BF$_4$]: 0.95 (3H, m, but-CH$_3$; C$_9$), 1.36 (2H, m, but-CH$_2$; C$_8$), 1.87 (2H, m, but-CH$_2$; C$_7$), 3.97 (3H, s, N-CH$_3$; C$_{10}$), 4.23 (2H, t, but-N-CH$_2$; C$_6$), 7.43 (1H, s, imidazole ring-N-CH-; C$_4$), 7.46 (1H, s, imidazole ring-N-CH-; C$_5$), 9.05 (1H, s, imidazole ring-N-CH-N; C$_2$); For n-Bu$_4$NPF$_6$: 0.90 (3H, m, but-CH$_3$; C$_4$), 1.318 (2H, m, but-CH$_2$; C$_3$), 1.83 (2H, m, but-CH$_2$; C$_2$), 3.224 (2H, t, but-N-CH$_2$; C$_1$); For MeCN: 2.021 (3H, s, CH$_3$-CN), & (b) $^1$H NMR (400 MHz, CDCl$_3$) spectrum of concentrated catholyte solution of CPBE experiment performed as per the reaction conditions given in 3rd row of Table 3: $\delta$ (ppm). No signals due to sodium formate, which appears at 9.6 ppm are seen. Scheme S1. A schematic diagram showing a plausible reaction mechanism proposed by Wang et al. [9], that is believed to takes place in an imidazolium based RTIL mediated ECR reaction to form CO in the presence of protons (H+) derived from water on the surface of an inert metal cathode.

**Funding:** This research was funded by SERB-DST (Government of India) grant number (No. SB/EMEQ-218/2014; dated: 17–06-2014).

**Acknowledgments:** Author also wishes to express his gratitude to all his colleagues at ARCI, Hyderabad, for their kind contributions to this study.

**Conflicts of Interest:** The author declares no conflict of interest.

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
