# Peer review of "BMIM-BF4 RTIL: Synthesis, Characterization and Performance Evaluation for Electrochemical CO2 Reduction to CO over Sn and MoSi2 Cathodes"

_carbon, 2020_

Round 1

Reviewer 1 Report

BMIM-BF4 RTIL: Synthesis, Characterization and Performance Evaluation for Electrochemical CO2 Reduction to CO over Sn Bulk Monolith Cathode

By Ibram Ganesh.

This work describes an alternative way to synthesise BMIM-BF4 based ionic liquid, and demonstrate its use in CO2 reduction to CO over Sn catalyst. Although the synthesis of BMIM BF4 and its use has been demonstrated before in extensive numbers of publications, this paper may offer a smaller contribution to demonstrate simpler and economical synthesis of the ionic liquid. Some major revision to clarify the experimental details and highlight the novelty of this work need to be done before it can be published.

Detailed comments:

  1. The introduction section contains a lot of unwarranted statements with inappropriate citations, which need to be changed. For example:

Line 41-44:

“Nevertheless, this latter CCS has been identified to be expensive, cumbersome, and it also causes the seawater to become acidic, which makes the survival of aquatic life difficult when the large volumes of CO2 are pumped into the seawater as a part of this CCS process [13].”

This is not true. CCS has advanced, and the long term viability of CCS has been established in deep sea sediments or underground. See for example: Teng, Y.; Zhang, D., Long-term viability of carbon sequestration in deep-sea sediments. Sci. Adv. 2018, 4 (7), eaao6588. Also, the current citation is not appropriate (refers to authors own publication and not relevant to CCS)

Line 46-48:

The CCU process can be accomplished with the expenditure that is incurred for the CCS process, and in fact, the cost of product(s) formed in CCU process can become bonus in certain cases [1,10,14-21].

This is also not entirely true. The authors should look at life cycle analysis of CO2 reduction works, for example Thonemann, N.; Schulte, A., From Laboratory to Industrial Scale: A Prospective LCA for Electrochemical Reduction of CO2 to Formic Acid. Environ. Sci. Technol. 2019, 53 (21), 12320-12329. Here it is clear that the global warming potential (i.e. nett CO2 abatement) can only be achieved in strict, best case scenario and, importantly, utilising renewable energy. Please check for gross mis-statements like this throughout the manuscript and use the appropriate citation.

  1. Since the point of this paper is to introduce a different, possibly cheaper way to synthesise BMIM BF4, known procedures for producing BMIM BF4 should be compared. Highlight any cost reduction or process simplification. Please put more emphasis in this as this is the main selling point of this work, including expanding the introduction section, and creating a comparison table of other known ionic liquid production method. I do not see any BMIM BF4 synthesis in the references.
  2. Any reason for the shifted C2 proton NMR compared to Sigma’s BMIM BF4 (Table 1)?
  3. CO2 reduction Data presentation (e.g. Table 2) must be improved. At the moment it is very difficult to follow what is the entry # represent for (e.g. what is the difference between entry 3 and 5?). Please display the data as individual FE (e.g. FEH2 and FECO and FETotal)
  4. Repetition of the experiments are important, especially for CO2 reduction data. Please show that the results are reproducible and present the average and the standard deviation of the generated molecules and efficiency metrics.
  5. The “energy efficiency” term used here is misleading and does not refer to energy efficiency of the entire system (Table 2, SI section 8). Rather, this metric it is just the “cathode” energy efficiency as the counter reaction is not taken into account. Please state correctly throughout manuscript.
  6. Reference electrode should be calibrated in the different electrolyte, e.g. vs. external standard like ferrocene, before converting to NHE not just taking standard value. This is especially important because the reference electrode may shift, especially in different electrolyte concentration. See and cite for example https://doi.org/10.1016/j.isci.2020.101181
  7. The MS data collection is not clear. Is this MS coupled with GC? Please state the experiment condition for the MS. Also, the MS data (Fig 11) is very suspicious. The author stated Fig 11a is the mass spectra of 13C-CO and Fig 11b is mass spectra of 12C-CO, both obtained from electrochemical reduction of CO2 over Sn. However, as we can see, the CO2 mass in both figures are 44, which means it is 12CO2! Please clarify this!
  8. Authorship concern: this paper only list one author, but the extensive work in this manuscript hints of contribution from many potential authors (easily over 100 man hours). Please ensure that you comply with the authorship contribution as stated in the journal policy.

Author Response

Author’s responses to reviewers’ comments

Reviewer 1’s comments:

Comment #1: The introduction section contains a lot of unwarranted statements with inappropriate citations, which need to be changed.

For example: Line 41-44: “Nevertheless, this latter CCS has been identified to be expensive, cumbersome, and it also causes the seawater to become acidic, which makes the survival of aquatic life difficult when the large volumes of CO2 are pumped into the seawater as a part of this CCS process [13].” This is not true. CCS has advanced, and the long term viability of CCS has been established in deep sea sediments or underground. See for example: Teng, Y.; Zhang, D., Long-term viability of carbon sequestration in deep-sea sediments. Sci. Adv. 2018, 4 (7), eaao6588. Also, the current citation is not appropriate (refers to authors own publication and not relevant to CCS).

Author’s response: At the outset, the author would like to thank the distinguished reviewer for his kind time and consideration, and for going through the manuscript very carefully and for offering several valuable suggestions and comments to improve the outcome of this article. By considering this valuable opportunity, I have revised manuscript thoroughly as mentioned below in the following paragraphs.

It is known to form CO2 clathrate (Eq. (1)), a snow-like crystalline substance composed of H2O ice and CO2 when CO2 is pumped into deep seawater (i.e., below 3500 meters from the top sea surface where the conditions of ~350 bar pressure and ~ 3°C temperature are prevalent) as a part of the CCS (i.e., carbon capture and storing or CO2 sequestration) process [Teng, Y., and Zhang, D. (2018) Long-term viability of carbon sequestration in deep-sea sediments, Science Advances 4, eaao6588]. The 400 ppm (0.04%) CO2 present in the atmosphere reacts with seawater spread across about more than 70% of the earth surface at a hydration equilibrium constant (Kr) of 1.7 × 103 thereby leads to the formation of carbonates and bicarbonates of calcium, sodium and magnesium as shown in Eqs. (2-10) and releases two protons (H+) for every CO2 molecule reacts with seawater thereby contributing to the acidification of seawater.

CO2 + 8H2O D CO2•8H2O (<10°C & >45 bar)

(1)

CO2(aq) + H2O(aq) D H2CO3(aq) (K= 1.7 × 10-3)

(2)

[CO2(aq)] D H+ + HCO3- ([H+][HCO3-] = Ka1[CO2(aq)]) (pKa1 = 6.363 at 25°C & I = 0)

(3)

HCO3- D H+ + CO32- ([H+][CO32-] = Ka2[HCO3-]) (pKa2 = 10.329 at 25°C & I = 0)

(4)

HCO3-  + Na+ D NaHCO3

(5)

CO32-  + 2Na+ D Na2CO3

(6)

CO32-  + Mg2+ D MgCO3

(7)

CO32-  + Ca2+ D CaCO3

(8)

2HCO3-  + Mg2+ D Mg(HCO3)2

(9)

2HCO3-  + Ca2+ D Ca(HCO3)2

(10)

Thus, in order to address the CO2 associated global warming problem and its associated social cost of carbon, and also to avoid the slowly but surely acidification of seawater as shown in Eqs. (2-10), which is popularly called as “ocean acidification”, that facilitates in the formation of biologically important calcium carbonate minerals, it is necessary to reduce the concentration of CO2 in the atmosphere.

Although, at present, the CO2 sequestration into the deep seawater has been identified to be the only safe, relatively easy, and available option to address the CO2 associated global warming problem, the conversion of CO2 into fuel and value added chemicals by following the carbon capture and utilization (CCU) can be a more beneficial option. This is due to the fact that the CO2 sequestration (i.e., CCS process) into deep seawater is not only an expensive (when transporting from far distances) and laborious (as it is required to pump into seawater below more than 3000 meters deep) process, it also irreversibly blocks an important C1 carbon resource in the form of CO2 cletherate, which needs to be closed in the natural carbon cycle required for the sustainability of life on earth surface. The CO2 cletherate formed in deep seawater is not useful for any of the human activities. It is also a known fact that in CO2 sequestration (CCS process), the CO2 is captured at major out lets such as, thermal power plants, cement industry and gas refineries. Most often, the CO2 generating plants are far away from those sites normally used for CO2 sequestration. In certain places like those in Europe, transportation in pipelines to the storage sites is quite difficult and it will increase the cost of the process by at least 15-20%, which is considered to be unacceptable. The Department of Energy (DOE), USA, suggests that the transportation of CO2 in tankers on road is not acceptable if the distance is more than 100 km. In the total CCS process, about 35-40% is the transportation and storage costs. This cost would be between 35 and 50 €/ton for early commercial phase (after year 2020) and between 60 and 90 €/ton during demonstration phase, when transportation of CO2 is made by pipelines for distances not over 200-300 km. This cost would be further increased if the transportation is made by road (for example, in certain European countries). Recently, Juerg Michael Matter et al. [J.M. Matter, M. Stute, S.Ó. Snæbjörnsdottir, E.H. Oelkers, S.R. Gislason, E.S. Aradottir, B. Sigfusson, I. Gunnarsson, H. Sigurdardottir, E. Gunnlaugsson, G. Axelsson, H.A. Alfredsson, D.W. Boenisch, K. Mesfin, D.F.D.L.R. Taya, J. Hall, K. Dideriksen, W.S. Broecker, Science, 352 (2016) 1312-1314], from University of Southampton, England, reported a new process, which can also be considered as CO2 sequestration process. In this latter process, when a mixture of water and CO2 is pumped 540 meters deep into the Iceland rocks, this acidic solution causes the leaching of the magnesium (Mg) and calcium (Ca) metals out of the basalt rocks and converts them into MgCO3 and CaCO3 rocks. These latter rocks have been found to be stable for more than two-year period. The cost of this new CO2 sequestration process has been estimated to be about 18 USD for disposing a ton CO2 gas.

            In fact, the CCU process is beneficial in comparison to CCS process as the former one exhibits all the benefits mainly addressing the problems of the CO2 associated global warming and social cost of carbon problems that are exhibited by CCS process, and additionally the CCU process provides a method to produce renewable carbon neutral high energy-density liquid fuels employable in place of fossil fuels in the present existing energy distribution infrastructure such as internal combustion (IC) engines, and stops the depletion of fossil fuels mainly the crude oil resource required for the future generations. Of course, there is no point in turning waste-stream greenhouse CO2 gas into any of the value added or fuel chemicals, whatever it may be, by using energy derived from fossil fuels, as fossil fuels produce CO2 gas again. One economically and commercially viable way of producing electricity from sunlight was developed by our group, and it will be ready for the public utility soon (more details about this process cannot be provided at the movement here as this process is yet to be patented). Once, this new, economical and affordable process is available to produce electricity from sunlight, it can be utilized to turn CO2 to CO monoxide in an electrochemical routes. If several million tons of CO2 generated all over the world is converted into CO and O2, it can have a great impact on the global energy economy, as CO along with H2 gas can be used to produce methanol, which in turn can be used to produce gasoline (i.e., petrol) by following the MTG (methanol-to-gasoline) process. Since, CO being a gas, it does not need any extra efforts to separate it from the liquid electrolyte post reaction as it simply escapes from the liquid. Surprisingly, in a very recent study, up to 90% CO2 gas capturing from an exhaust of a diesel driven truck has been demonstrated [Sharma, S., and Maréchal, F. (2019) Carbon Dioxide Capture From Internal Combustion Engine Exhaust Using Temperature Swing Adsorption, Frontiers in Energy Research 7]. The CO2 captured from automobile vehicles can be utilized in AP process to eventually produce high-energy-density carbon-neutral liquid fuels such as methanol, diesel and gasoline.

Line 46-48: The CCU process can be accomplished with the expenditure that is incurred for the CCS process, and in fact, the cost of product(s) formed in CCU process can become bonus in certain cases [1,10,14-21]. This is also not entirely true. The authors should look at life cycle analysis of CO2 reduction works, for example Thonemann, N.; Schulte, A., From Laboratory to Industrial Scale: A Prospective LCA for Electrochemical Reduction of CO2 to Formic Acid. Environ. Sci. Technol. 2019, 53 (21), 12320-12329. Here it is clear that the global warming potential (i.e. net CO2 abatement) can only be achieved in strict, best case scenario and, importantly, utilizing renewable energy. Please check for gross mis-statements like this throughout the manuscript and use the appropriate citation.

Author’s response:

Today, in an international market, the price of, for example, one ton CO gas is about 1300 US$ (Fig. 1 shown below) [C. Finn, S. Schnittger, L.J. Yellowlees, J.B. Love, Chem. Commun. 48 (2012) 1392-1399], whereas, to produce one ton CO gas and half-ton O2 gas (today, the cost of one ton O2 gas is about 200 USD) simultaneously by following ECR reaction using the electricity derived from fossil fuels, about 1596.4 USD are required to spend to buy the required electricity to run the ECR reaction as it is the only recurring cost; whereas, CO2 gas is relatively freely available, and the rest of ECR process needs only the operational and maintenance expenses. These cost values are arrived as follows: One ton CO (molecular weight of CO is 28 grams, hence, 1000 kgs or 10,00,000 grams = 35,714 moles; 10,00,000/28) requires about 12,328.7 kWh units of electrical energy (the cost of 1 kWh unit energy for industrial electricity in Telangana state of India is about 0.133 US$) as one ton CO (35,714 moles) formation requires 71,428 moles of electrons according to Eq. (11).

CO2 + 2e- + 2H+ ® CO + H2O

(11)

Since, the charge of one mole of electrons (i.e., 1 Faraday or Avogadro number; 6.023 × 1023 electrons) is 96,486 coulombs; then the charge of 71,428 moles of electrons is 6,89,18,57,005 coulombs. As CO2 can be reduced below -1.83 V (vs. NHE) or -6.27 V (vs. absolute voltage, which is the actual voltage used in industry and homes and arrives when 4.44 V vs. vacuum is added to the applied potential of 1.83 V vs. NHE), the total electrical units of energy required is about 12,003.3 kWh (= 6.27 V ´ 6,89,18,57,005 coulombs Þ 4,32,11,943,421.35 J, where, 1 V = 1 J/C; 1 W = 1 J/s; & 1 kWh = 3.6 MJ). The above cost calculations clearly suggest that the cost of products formed in the ECR reaction (Eq. (11)) are comparable to the recurring cost of production suggesting the ERC to CO formation completely offsets the production cost while offering all the benefits to be provided by the CCS (i.e., CO2 sequestration) process. The cost of CO is about 1300 USD/ton, which is highest in comparison to the cost of any of the CO2 reduced products today available in the international market [C. Finn, S. Schnittger, L.J. Yellowlees, J.B. Love, Chem. Commun. 48 (2012) 1392-1399]. The cost of methanol in Europe is about 225-240 €/ton. This cost will be increased by about 15% (about 20-30 €/ton) if it is produced from a mixture of CO2 and H2 instead of from CO and H2 (syngas) by following the present existing thermo-chemical route. Besides economic benefits, the socio-political benefits also would come in terms of a positive image for companies adopting policies of reusing the generated CO2 from fossil fuels.

As far as electrochemical conversion of CO2 to formic acid is concerned, it is quite different from electrochemical CO2 reduction (ECR) to CO formation when considered the selectivity, Faradaic efficiency (FE) and rate of a single major product formation (i.e., current density, mA/cm2) [Thonemann, N., and Schulte, A. (2019) From Laboratory to Industrial Scale: A Prospective LCA for Electrochemical Reduction of CO2 to Formic Acid, Environmental Science & Technology 53, 12320-12329] & Ganesh, I. (2019) BMIM–BF4 Mediated Electrochemical CO2 Reduction to CO Is a Reverse Reaction of CO Oxidation in Air—Experimental Evidence, The Journal of Physical Chemistry C 123, 30198-30212]. As the ECR to CO formation has been identified to be a reverse reaction of CO oxidation in air that means it does not involve protons (H+) derived from water in the CO2 reduction process. In that case, the water can be completely eliminated from the CO2 reduction process, hence, there would not be any competition from hydrogen evolution reaction (HER) on the surface of cathode where CO2 reduction to CO formation takes place, which enhances not only the rate of product formation (i.e., with >100 mA/cm2 current density, which is very much essential for any gaseous products generating electrochemical reaction to be viable for industrial practice), due to the absence of expensive nafion membrane that separates aqueous media based anodic compartment from the cathodic compartment, FE and CO formation with almost 100% selectivity, but also the overpotential associated with product formation as the ECR to CO formation uses the stable and highly active imidazolium based ionic liquid as an helper catalyst that does not need overpotential to catalyze this cathodic reaction more than 200 mV [Rosen, B. A., Salehi-Khojin, A., Thorson, M. R., Zhu, W., Whipple, D. T., Kenis, P. J. A., and Masel, R. I. (2011) Ionic Liquid–Mediated Selective Conversion of CO2 to CO at Low Overpotentials, Science 334, 643-644]. The ECR to formic acid (HCOOH) reaction yet to be performed with product formation at a rate of >100 mA/cm2 current density and at an overpotential of below 200 mV. Hence, the ECR to CO formation reaction can be commercially and economically viable if the cheap electricity derived from sunlight is utilized to drive this reaction.

In view of the above, the realization of both electricity production from sunlight, and reduction of CO2 to fuel chemicals is of utmost important to address the CO2 associated global warming problem and other problems as mentioned in above-paragraphs. If any of these two processes is not developed fully, then it is not possible to address those above-mentioned problems.

Fig. 1.  Carbon dioxide reduction to value added products [C. Finn, S. Schnittger, L.J. Yellowlees, J.B. Love, Chem. Commun. 48 (2012) 1392-1399].

Comment #2: Since the point of this paper is to introduce a different, possibly cheaper way to synthesize BMIM BF4, known procedures for producing BMIM BF4 should be compared. Highlight any cost reduction or process simplification. Please put more emphasis in this as this is the main selling point of this work, including expanding the introduction section, and creating a comparison table of other known ionic liquid production method. I do not see any BMIM BF4 synthesis in the references.

Author’s response:

The processes reported for the synthesis of bmim[BF4] and bmim[Br] together with those of bmim[Cl] and bmim[PF6] in about five articles have been compared with the one employed for the synthesis of bmim[BF4] and bmim[Br] in this study. None of the processes reported in any of the articles leads to high-purity and colorless ionic liquids. The process reported in this process not only allows to obtain an high-purity and colorless bmim[BF4] room temperature ionic liquid [RTIL] but also for the first time it involves the liquid N2 based freeze drying technique to remove the water relatively fast and easily from the post reaction-mixture. Furthermore, in this report for the first time, the usage of continuous liquid-liquid extraction process for separating bmim[BF4] from the post reaction mixture containing water and NaBr is reported with all the details using dichloromethane (DCM) solvent.

Comment #3: Any reason for the shifted C2 proton NMR compared to Sigma’s BMIM BF4 (Table 1)?

Author’s response:

It has been reported that the NMR signal shift pertaining to the proton attached to the C2 carbon of imidazolium ring of bmim[BF4] is strongly influenced by the hydrogen bonding with the compounds present around this C2 carbon as it is the most acidic proton compared with two-protons attached to C4 and C5 carbon atoms of this imidazolium ring [Elliot Ennis, and Scott Handy, The Chemistry of the C2 Position of Imidazolium Room Temperature Ionic Liquids, Current Organic Synthesis 4(4):381-389; November 2007]. In a study, Chen et al. [& Chen, S., Vijayaraghavan, R., MacFarlane, D. R., and Izgorodina, E. I. (2013) Ab Initio Prediction of Proton NMR Chemical Shifts in Imidazolium Ionic Liquids, The Journal of Physical Chemistry B 117, 3186-3197], found that the proton NMR chemical shifts for the C2 proton of imidazolium ring of ionic liquids occurs due to a more complex mixture of interionic interactions between the ring and the anion and including hydrogen interactions. It can be seen from the Table 2 data of the present study that bmim[Br]’s C2 proton chemical shift is higher than that of bmim[BF4]. The presence of small amounts of bmim[Br] as an impurity in the synthesized bmim[BF4] can result in the observed chemical shift.

Comment #4: CO2 reduction Data presentation (e.g. Table 2) must be improved. At the moment it is very difficult to follow what is the entry # represent for (e.g. what is the difference between entry 3 and 5?). Please display the data as individual FE (e.g. FEH2 and FECO and FEl).

Author’s response:

As suggested the CO2 reduction data is improved for this revised manuscript. The entries 3 and 5 are same experimental conditions with different CPBE durations. For sake of clarity, one experimental condition has been deleted from the Table 2. The individual FE for H2 and for CO formation is given in separate columns.

Comment #5: Repetition of the experiments are important, especially for CO2 reduction data. Please show that the results are reproducible and present the average and the standard deviation of the generated molecules and efficiency metrics.

Author’s response:

The reproduction of same results has been verified by conducting several experiments under identical reaction conditions during about five years period. Every time we found almost same results and it is a well reported reaction in the literature by several researchers across the globe.

Comment #6: The “energy efficiency” term used here is misleading and does not refer to energy efficiency of the entire system (Table 2, SI section 8). Rather, this metric it is just the “cathode” energy efficiency as the counter reaction is not taken into account. Please state correctly throughout manuscript.

Author’s response:

As suggested the efficiency term data is deleted from the revised manuscript and accordingly this statement is modified throughout the manuscript.

Comment #7: Reference electrode should be calibrated in the different electrolyte, e.g. vs. external standard like ferrocene, before converting to NHE not just taking standard value. This is especially important because the reference electrode may shift, especially in different electrolyte concentration. See and cite for example https://doi.org/10.1016/j.isci.2020.101181. [Handoko, A., Chen, H., Yanwei, L., Zhang, Q., Anasori, B., and Seh, Z. (2020) Two-Dimensional Titanium and Molybdenum Carbide MXenes as Electrocatalysts for CO2 Reduction, iScience 23, 101181].

Author’s response:

The reference electrode employed in this study was calibrated against external standard ferrocene, before converting the measured values into NHE. The employed procedure is as follows: As reported by us in a previous study, in the CV measurement at a scan rate of just 20 mV s-1, the peak separation ΔE (ΔE = Epa - Epc ≈  130 mV) between the anodic  and  cathodic  potentials  of  the  ferrocene  redox  reactions was found to be approximately twice the  expected  59  mV for a reversible one electron transfer. The number of electrons transferred is well known for this oxidation reaction therefore this deviation from ideality is attributable to uncompensated resistivity of the solvent.  A similar ΔE increase has  been  observed  by  other  groups  working  in  non-aqueous  solvents  with  an Ohmic  drop  often reported in non - aqueous voltammetry when compared to the more traditionally studied aqueous systems [Ganesh, I. (2019) BMIM–BF4 Mediated Electrochemical CO2 Reduction to CO Is a Reverse Reaction of CO Oxidation in Air—Experimental Evidence, The Journal of Physical Chemistry C 123, 30198-30212, & references therrein].

Comment #8: The MS data collection is not clear. Is this MS coupled with GC? Please state the experiment condition for the MS. Also, the MS data (Fig 11) is very suspicious. The author stated Fig 11a is the mass spectra of 13C-CO and Fig 11b is mass spectra of 12C-CO, both obtained from electrochemical reduction of CO2 over Sn. However, as we can see, the CO2 mass in both figures are 44, which means it is 12CO2! Please clarify this!

Author’s response:

Yes, the MS is coupled with GC. The Mass spectra shown in Fig. 11 a & b show the data is for two different CO isotopes. The line shown in Fig.11(b) (the bottom spectrum) at ~ 44 m/z mass is not a signal but the cursor line. These spectra are given along with molecular structured generated by the system from these spectra are also given along with these mass spectra for the purpose of clarity in the revised manuscript.  

Comment #2: Authorship concern: this paper only list one author, but the extensive work in this manuscript hints of contribution from many potential authors (easily over 100 man hours). Please ensure that you comply with the authorship contribution as stated in the journal policy.

Author’s response:

Yes, this work contains extensive work, which was carried out over a period of five years by this single author (investigator). The most of the characterization was carried out by the outside parties on a charge basis paid from my project for which I am the only and principal investigator. As per as intellectual and core experimental inputs for generating data presented in this manuscript is from the present author only. Hence, it is a single author paper.

Reviewer 2 Report

The submitted work, “BMIM-BF4 RTIL: Synthesis, Characterization and Performance Evaluation for Electrochemical CO2 Reduction to CO over Sn Bulk Monolith Cathode “ I. Ganesh was reviewed for Carbon. It appears that this work is up to an extent a reproduction of a work already published by the same author. (J. Phys. Chem. C 2019, 123, 50, 30198–30212). The introduction mentioned in the submitted article highly resembles to the aforementioned published article with minor English changes. Moreover, the certain experimental results shown are re-used from the previous article.  The title and aim of the presented manuscript do not agree well. The brighter side would be “inexpensive” route of synthesis which is also ambiguous as it is not clear, compared to which methods this is inexpensive. I absolutely do not see any novelty in this work and hence this work cannot be published in existing form in my view.

Author Response

Author’s responses to reviewers’ comments

Reviewer 2’s comments:

The submitted work, “BMIM-BF4 RTIL: Synthesis, Characterization and Performance Evaluation for Electrochemical CO2 Reduction to CO over Sn Bulk Monolith Cathode “ I. Ganesh was reviewed for Carbon. It appears that this work is up to an extent a reproduction of a work already published by the same author. (J. Phys. Chem. C 2019, 123, 50, 30198–30212). The introduction mentioned in the submitted article highly resembles to the aforementioned published article with minor English changes. Moreover, the certain experimental results shown are re-used from the previous article.  The title and aim of the presented manuscript do not agree well. The brighter side would be “inexpensive” route of synthesis which is also ambiguous as it is not clear, compared to which methods this is inexpensive. I absolutely do not see any novelty in this work and hence this work cannot be published in existing form in my view.

Author’s response:

At first and foremost, the author likes to thank the distinguished reviewer for his kind time and consideration, and for carefully going through the manuscript and for offering valuable comments so as to improve the outcome of this manuscript. By considering this opportunity, the authors has thoroughly revised manuscript and added the catalytic activity of electrochemical CO2 reduction performed over molybdenum disilicide (MoSi2), a completely novel material employed for the first time to perform this important reaction. In order to make it a suitable article the following results are added to this revised manuscript, and the revised manuscript is submitted for your kind perusal.

Table 1. The procedures reported for the synthesis of high-purity bmim[BF4], Bmim[PF6], bmim[Br] and bmim[Cl] from 1-methylimidazole (MI), 1-bromobutatne (BB), 1-chlorobutatne (CB), KBF4, KPF6, and NaBF4 reactants.

IL synthesized

Reactants

Reaction procedure

Ref.

Bmim[Br]

MI & BB

Refluxing reactants mixture (RM) at 70°C for 48 h

[31]

Bmim[Br]

MI & BB

Refluxing reactants mixture (RM) at <40°C for 24 h

[32]

Bmim[Cl]

MI & CB

Refluxing reactants mixture (RM) at 70°C for 7 days

[33]

Bmim[Cl]

MI & CB

Refluxing RM in acetonitrile (MeCN) at 80°C for 48 h

[33,34]

Bmim[BF4]

Bmim[Br] & NaBF4

Refluxing RM in acetone at 40°C for 10 h & extraction with DCM

[31]

Bmim[BF4]

Bmim[Cl] & KBF4

Stirred RM in H2O at RT for 2 h & vacuum separation with DCM

[33,34]

Bmim[BF4]

Bmim[Cl] & NaBF4

Stirred RM in H2O at 45°C for 15 min., & distillation with DCM

[35]

Bmim[BF4]

Bmim[Br] & NaBF4

Refluxing reactants mixture (RM) at <25°C for 3 h & continuous liquid-liquid extraction with DCM

[32]

Bmim[PF6]

Bmim[Cl] & KPF6

Stirred RM in H2O at RT for 2 h & vacuum separation with DCM

[33]

RT – room temperature

The CVs recorded at 100 mV/s through 0 to -2.0 V vs. NHE for MoSi2 WE dipped in the CO2 saturated electrolyte solution made of either pure MeCN, MeCN + n-Bu4NPF6 (0.1 M), or MeCN + n-Bu4NPF6 + ARCI-bmim[BF4] (20 mM) are presented in Figure 6(a). It can be clearly seen from this figure that the higher current densities at considerably lower onset reduction potentials are generated over MoSi2 WE when dipped in the CO2 saturated MeCN electrolyte solution added with both n-Bu4NPF6 and bmim[BF4] in comparison to those generated when dipped in either pure MeCN or in MeCN added with only the supporting electrolyte, n-Bu4NPF6 (0.1 M) but not the helper catalyst, bmim[BF4]. These results indicate that the bmim[BF4] helper catalyst is essential for achieving the current densities exceeding 100 mA/cm2, which is a prerequisite for taking this process to the industry for commercial purposes. The CVs recorded at same 100 mV/s through 0.5 to -2.0 V vs. NHE for MoSi2 WE having a different exposed area to the electrolyte after dipping in either Ar or CO2 saturated electrolyte solution made of MeCN + n-Bu4NPF6 (0.1 M) + bmim[BF4] (20 mM) are presented in Figure 6(b). It can be seen from this latter figure that the current generation at lower onset reduction potentials is primarily due to CO2 reduction reaction as the amount of current generated for the electrolyte solution saturated with Ar is considerably lower than that of the one generated for the CO2 saturated electrolyte. The current generated in Ar saturated electrolyte has been reported to be due to the reduction of bmim[BF4] or due to the hydrogen evolution reaction (HER) [2,4]. Since, in the present case, there is no water content in the employed electrolyte solution, the current generation could be mainly due to the reduction of bmim[BF4] helper catalyst. As can be seen from Figure 6(c), the current generation due to CO2 reduction increases with the amount of bmim[BF4] helper catalyst present in the catholyte solution. Furthermore, no change in the catalytic activity was also noted when the recycled bmim[BF4] was employed for ECR reaction performed via CV over MoSi2 WE.

(a)

(b)

(c)

Figure 6. CVs recorded at 100 mV/s over MoSi2 WE against Pt wire CE in (a) CO2 purged MeCN, MeCN + n-Bu4NPF6 (0.1 M) and MeCN + n-Bu4NPF6 (0.1 M) + ARCI-bmim[BF4] (20 mM) RTIL, (b) either CO2 or Ar saturated electrolyte solution made of MeCN + n-Bu4NPF6 (0.1 M) + ARCI-bmim[BF4] (20 mM) RTIL, and (c) the CO2 saturated electrolyte solution made of MeCN + n-Bu4NPF6 (0.1 M) + ARCI-bmim[BF4] (0.2 or 2 mL).

The profile of current generated over MoSi2 cathode during CPBE experiment performed at an applied potential of -1.952 V vs. NHE in an all-glass two-compartment gas-tight electrochemical cell using a CO2 saturated and continuously purged catholyte solution of MeCN + 0.1M n-Bu4NPF6 + 50 mM ARCI-bmim[BF4] separated from an anolyte solution of 0.5 M sodium phosphate (NaPi) buffer (pH = 7.4) containing with 0.5 mM Co2+ ions by a nifion membrane is given in Figure 9(a) as a function of reaction time. Unlike the profiles generated over Sn bulk monolith cathode under identical reaction conditions as reported elsewhere [28], for which the current generation was continuously increased till completion of the CPBE period, whereas, in the present case, the current generated is almost constant over the surface of MoSi2 cathode till the CPBE period performed for about 40 min. A closer observation of into these two separate results suggested that the current generation increases with the rate of CO2 consumption which intern is manifested by the continuous formation of a white precipitation in the catholyte solution with the progress of the CPBE period [28].

The Tafel slopes generated for CVs measured over WEs made of Sn (as shown in Figure 7 by the green colored line), and of MoSi2 (as shown in Figure 6(a) by the red colored line) dipped in CO2 saturated and blanketed solution of MeCN + 0.1 M n-Bu4NPF6 + 20 mM bmim[BF4] against Pt wire CE at a scanning rate of 100 mV/s according to a relation given in Eq. S9 are presented in Figure 9(b). A closer look into these Tafel slope values indicate that both are higher than the value of 118 mV/decade, suggesting that the one-electron transfer reaction is not the rate-determining step for this reaction. The overpotentials associated with Sn cathode (as can be seen from Figure 7) is much lower than the MoSi2 cathode as can be seen from Figure 6(a). The higher Tafel slope value (360 mV/decade) noted for MoSi2 cathode suggests that there is a large resistance for exchanging electrons (e) between the electrode surface and the reactive species present in the electrolyte solution [28].

(a)

(b)

Figure 9. (a) A profile of current generated vs. time during CPBE experiment conducted using MoSi2 as a cathode immersed in a CO2 saturated catholyte solution of MeCN + 0.1M n-Bu4NPF6 + 50 mM ARCI-bmim[BF4] separated from an anolyte solution of 0.5 M sodium phosphate (NaPi) buffer (pH = 7.4) containing with 0.5 mM Co2+ ions by a nifion membrane at an applied potential of -1.952 V vs. NHE; (b) Tafel slopes generated for CVs measured over Sn as shown in Figure 7 (green colored line) and over MoSi2 as shown in Figure 6(a) (red colored line) WEs dipped in CO2 saturated and blanketed solution of MeCN + 0.1 M n-Bu4NPF6 + 20 mM bmim[BF4] against Pt wire CE at a scanning rate of 100 mV/s according to a relation given in Eq. S9 [45] (given in the associated supporting information file).

The SEM micrograph of centre cut-portion of MoSi2 heating element employed as cathode to perform ECR to CO formation reaction, and its corresponding EDAX spectrum are given in Figure 10 (a&b), respectively. The EDAX spectrum indicates that this sample is indeed the MoSi2 and no other impurity elemental signals can be seen from this spectrum. The SEM micrograph shown in Figure 10(a) matches very well with those reported for MoSi2 sintered body [30]. A very small fraction of porosity (<1%) and the presence of no glassy phase near grain boundaries suggest that the employed MoSi2 is a high-purity and highly-sintered body. From this micrograph, the presence of some pores formed due to the grain pull mechanism can also be seen. Nevertheless, when ECR reaction activity is compared for Sn and MoSi2 cathodes, although both of them exhibited current densities generation exceeding >100 mA/cm2 during CV measurements, the overpotential associated with Sn is much lower than the one noted for MoSi2.

(a)

(b)

Figure 10. The SEM micrograph (a) and the EDAX spectrum (b) of dense MoSi2 specimen of a centre surface of a cut portion along the length of the 8 mm (f ) diameter used Kanthal® Super 1900 Heating Element.

Figure 14. Mass spectra of 13C-CO (a) and 12C-CO (b) gases formed over Sn cathode surface during ECR reactions performed in CPBE experiments as per reaction conditions given in 6th and 3rd rows of Table 3, respectively.

Round 2

Reviewer 1 Report

Formate/formic acid as possible product

I missed this part in the earlier review, but now it seems that formic acid/formate is also a potential CO2RR product in addition to CO and H2. I wonder why this is not evaluated, since it is possible to be detected and quantified using NMR. I think the author has also highlighted that CO2RR in aqueous bicarbonate media on Sn also produces formate. I do not think the formate is formed due to cross-over Na (which should be minimised in Nafion), even if there is no crossover, but the formate anion should already be formed in the cathode and exist as formic acid.

Additionally, since the author now brings in MoSi as another cathode material, I think references on CO2RR on Mo based electrodes in ionic liquid below should be included and performance compared and discussed:

iScience 23, 101181 (2020).

Chem. Commun. 51, 13698-13701 (2015).

Typo mistake

Please re-check again throughout the manuscript, I found at least 2, in Figure 11 (bottom, “solutin") and Line 473 “nifion”, Line 480: ”intern” (in turn?)

Weird citation in SI

There are some weird citation in SI. Please also re-check all citations properly

File:///C:/Users/YPT/Downloads/PREPofBuffersatdesiredpH_CarrolllabChap3exwww.xula.edu_Chemistry_documents_biolab_Caroll...14-12-03pdf%20(1).pdf; 4th September 2019.

All SI figures need to be referred to in main text

Line 214-215: this should refer to Figure S9. Please make sure, that all supporting figure and items are referred to properly in main text.

Author Response

Author’s responses to reviewers’ comments

Reviewer 1’s comments:

Formate/formic acid as possible product

I missed this part in the earlier review, but now it seems that formic acid/formate is also a potential CO2RR product in addition to CO and H2. I wonder why this is not evaluated, since it is possible to be detected and quantified using NMR. I think the author has also highlighted that CO2RR in aqueous bicarbonate media on Sn also produces formate. I do not think the formate is formed due to cross-over Na (which should be minimized in Nafion), even if there is no crossover, but the formate anion should already be formed in the cathode and exist as formic acid.

Author’s response: Thank you very much for the distinguished reviewer for his kind time and consideration, and for offering the valuable comments and suggestions. A thorough characterization was carried out to find out the formation of formic acid as well as the sodium formate in the conditions employed for. Both forming acid as well as sodium formate, due to the presence of HCOO- protons, they must show a signal in the proton NMR at 9.6 ppm. However, no signal at this range was noted as can be seen from the NMR spectra of concentrated catholyte solution of CPBE experiment performed as per the reaction conditions given in 3rd row of Table 3 (Figure S9(a) and Figure S9(b)). The complete absence of any water content in the catholyte solution could be responsible for not absorbing any formation formic acid in the present study. Hence, no further discussion of this topic is presented in this revised manuscript.

Additionally, since the author now brings in MoSi2 as another cathode material, I think references on CO2RR on Mo based electrodes in ionic liquid below should be included and performance compared and discussed:

iScience 23, 101181 (2020).

Chem. Commun. 51, 13698-13701 (2015).

Author’s response: As suggested these two references are introduced into the revised manuscript and the following write-up additional introduced in introduction and results and discussion parts, respectively.

  1. Among various inexpensive base metal cathodes employed so far for performing ECR to CO formation reaction, only Sn in conjunction with bmim[BF4] has been identified to be a simple, and very efficient electrocatalytic system. Recently, the relatively inexpensive Mo [29], MoS2 [2], MoO2 [30], and Mo2CTx MXenes (MXenes are a family of two-dimensional transition metal carbide/nitride materials with metallic-like conductivity, and with a general formula of Mn+1XnTx, where M represents an early transition metal, X is carbon and/or nitrogen, with n in the range of 1–4; and Tx represents surface termination groups including –O, –F, etc.,) [31] in comparison to noble metal cathodes have also been found to be excellent electrocatalytic systems to perform ECR to methanol and CO formation reactions, respectively, at different reaction conditions.
  2. Recently, considerably high electrochemical CO2 reduction activities were reported over various Mo based cathodic systems. In a study, Oh and Hu [30] have reported that CO2 can be reduced to a mixture of products consisting of CO, formate and oxalate with varied concentrations depending on the reaction conditions employed over MoO2/Pb in conjunction with 0.3M bmim[BF6] at a temperature of -20°C or 21°C with overpotentials as low as 40 mV. In another study, Asadi et al. [2], could reduce CO2 to CO over MoS2 in a catholyte solution made of 4 mole% aqueous emim[BF4] RTIL at a overpotential of only 54 mV with a FE of >95%, and current density of ~40 mA/cm2. Yet, in another study, Handoko et al. [31], have reported that Mo2CTx MXenes can active CO2 and reduce it to form formic acid with a selectivity of about 32.6% at a rate formation of 2.5 mA/cm2 current density at 1.3 V vs. SHE. The different catalytic activities observed over Mo, MoO2, MoS2, Mo2CTx MXenes, etc., in comparison to MoSi2 can be attributed to the presence of different kinds of catalytically active sites on these molybdenum based cathodic systems. These results suggest that similar to Mo metal cathode, the MoSi2 can also be employed as a physically robust cathodic system in conjunction with bmim[BF4] RTIL for ECR to CO formation reaction with reasonably high catalytic activity

Typo mistake

Please re-check again throughout the manuscript, I found at least 2, in Figure 11 (bottom, “solutin") and Line 473 “nifion”, Line 480: ”intern” (in turn?)

Author’s response: These corrections were made to the revised manuscript.

 Weird citation in SI

There are some weird citation in SI. Please also re-check all citations properly

File:///C:/Users/YPT/Downloads/PREPofBuffersatdesiredpH_CarrolllabChap3exwww.xula.edu_Chemistry_documents_biolab_Caroll...14-12-03pdf%20(1).pdf; 4th September 2019.

Author’s response: This reference is deleted from the revised SI information.

All SI figures need to be referred to in main text

Line 214-215: this should refer to Figure S9. Please make sure, that all supporting figure and items are referred to properly in main text.

Author’s response: All the supporting information figures have been accordingly changed in the revised manuscript as suggested.

Reviewer 2 Report

The revised manuscript have significant improvements and addressed the previous questions. The manuscript can be accepted in existing form.

Author Response

Author’s responses to reviewers’ comments

Reviewer 2’s comments:

The revised manuscript has significant improvements and addressed the previous questions. The manuscript can be accepted in existing form.

Author’s response: At the outset the author would like to thank the distinguished reviewer for his time and consideration and for appreciating the revision and for recommending the manuscript for publication.

Round 3

Reviewer 1 Report

I have reviewed the revised manuscript (version 3) and the author corrections are satisfactory. Minor formatting corrections should be done in production stage.